



# Optimal Estimation Retrieval Framework for Daytime Clear-Sky Total Column Water Vapour from MTG-FCI Near-Infrared Measurements

Jan El Kassar[1,3], Cintia Carbajal Henken[1], Xavier Calbet[2], Pilar Rípodas[2], Rene Preusker[1], and Jürgen Fischer[1,3]

[1]Institute of Meteorology, Freie Universität Berlin, Carl-Heinrich-Becker-Weg 6-10, 12165 Berlin, Germany
[2]Agencia Estatal de Meteorología, Leonardo Prieto Castro 8, Ciudad Universitaria, 28071 Madrid, Spain
[3]Spectral Earth GmbH, Baseler Str. 91a, 12205 Berlin, Germany

**Correspondence:** Jan El Kassar, jan.elkassar@met.fu-berlin.de; Tel.: +49-30-838-64876

**Abstract.** A retrieval of total column water vapour (TCWV) from the new daytime, clear-sky near-infrared measurements of the Flexible Combined Imager (FCI) on-board the geostationary satellite Meteosat Third Generation Imager (MTG-I, Meteosat-12) is presented. The retrieval algorithm is based on the differential absorption technique, relating TCWV amounts to the radiance ratio of a non-absorbing band at 0.865 $\mu$m and a nearby water vapour (WV) absorbing band at 0.914 $\mu$m. The sensitivity of the

band ratio to WV amount increases towards the surface, which means the whole atmospheric column down to the boundary layer moisture variability can be observed well.

The retrieval framework is based on an Optimal Estimation (OE) method providing pixel-based uncertainty estimates. It builds on well-established algorithms successfully applied to other passive imagers with similar spectral band settings. Transferring knowledge gained in their development onto FCI required some new approaches. The absence of additional, adjacent

window bands to estimate the surface reflectance within FCI's absorbing channel were mitigated using a Principle Component Regression (PCR) from the bands at 0.51, 0.64, 0.865, 1.61 and 2.25 $\mu$m.

Since a long-term calibrated FCI dataset is not available yet, we build a second forward model for two equivalent NIR bands (0.865 and 0.9 $\mu$m) on the Sentinel-3 Ocean and Land Colour Instrument (OLCI). A long-term validation of OLCI against a single Atmospheric Radiation Measurement (ARM) reference site without the PCR resulted in a bias of 1.85 kg/m$^2$,

cRMSD of 1.26 kg/m$^2$ and $r^2$ of 0.995. In order to test the PCR which uses FCI bands in the visible to short-wave infrared, we replaced the bands missing in OLCI with bands from the Sea and Land Surface Temperature Radiometer (SLSTR). A spectrally similar dataset was created from SLSTR and OLCI data on Sentinel-3A/B during June 2021. This dataset is used to test the retrieval with regards to robustness and global performance of the PCR. A first verification of this OLCI/SLSTR "FCI-alike" TCWV against well-established ground-based TCWV products concludes with a wet bias between 1.23 – 3.12

kg/m$^2$, a cRMSD between 1.88 – 2.35 kg/m$^2$ and $r^2$ between 0.95 – 0.97. In this set of comparison, only land pixels were considered. Furthermore, a dataset of FCI Level 1c observations with a preliminary calibration was processed. The TCWV processed from FCI data aligns well with reanalysis TCWV and collocated OLCI/SLSTR TCWV but show a dry bias. A more rigorous validation and assessment will be done, once a longer record of FCI data is available.



The PCR may be extended to include more diverse water-bodies. In future iterations, more bands in the visible spectral range may be added to further increase performance in presence of aerosol over dark surfaces.

This novel TCWV dataset derived from geostationary satellite observations enhances monitoring of WV distributions and associated meteorological phenomena from synoptic scales down to local scales. Such observations are of special interest for the advancement of nowcasting techniques and Numerical Weather Prediction (NWP) accuracy as well as process-studies.

## 1 Introduction

The new Meteosat Third Generation Imager (MTG-I, hereinafter referred to as MTG) carries the Flexible Combined Imager (FCI) (Holmlund et al., 2021; Martin et al., 2021). This third generation of European geostationary meteorological satellites is commissioned by the European Organisation for the Exploitation of Meteorological Satellites (EUMETSAT) for monitoring weather and climate. FCI is the successor to the Spinning Enhanced Visible and Infrared Imager (SEVIRI) (Schmetz et al., 2002) and will both enhance the temporal and spatial resolution of geostationary remote sensing observations. Also, an expanded set of spectral channels compared to SEVIRI allows for more comprehensive observations of atmospheric and surface properties. Even more compelling is, that the instrument FCI which measures radiances across the visible (VIS), near-, shortwave-, midwave- and longwave-infrared spectra, includes a new band not available on any other instrument onboard a geostationary platform to date. This band is located within one of the $\rho\sigma\tau$ water vapour (WV) absorption regions in the Near-Infrared (NIR) at 0.913 $\mu$m. There, light is more likely to be absorbed by WV molecules compared to spectral regions outside these absorption features (window regions). Moreover, these NIR measurements exhibit the greatest sensitivity to WV amounts near the surface. Consequently, this allows for probing the entire atmospheric column, thus the retrieval of accurate clear-sky total column water vapour (TCWV) fields as well as providing information on changes of WV amounts in the lower troposphere. This contrasts with WV retrievals from thermal infrared measurements. On the one hand, a split-window technique using weakly absorbing WV measurements can be employed to retrieve TCWV or boundary layer WV with relatively high uncertainties (e.g., Kleespies and McMillin, 1990; Hu et al., 2019; Dostalek et al., 2021; El Kassar et al., 2021). Lindsey et al. (2014) and Lindsey et al. (2018) showed that the split-window difference by itself may already provide valuable insight on the WV content in the boundary layer or lowest layers of the troposphere. On the other hand, measurements from strongly absorbing WV bands serve to retrieve WV amounts limited to upper tropospheric levels and/or layered WV products (e.g., Koenig and De Coning, 2009; Martinez et al., 2022).

The introduction of MTG and its new FCI NIR band, particularly in combination with increased temporal coverage of 10 minutes, greatly expands our ability to quantify and characterize local to global scale water vapour distributions and monitor their changes. This has important implications for both weather and climate research and applications. Atmospheric WV is the fundamental ingredient in the formation of clouds and precipitation. Spatio-temporal WV distributions and fluxes impact intensity and duration of precipitation processes. The presence of sufficient low-level moisture in the atmospheric boundary layer facilitates the formation of convective development through enhancement of atmospheric instability and also contributes



to storm severity by acting as a source of energy, once a storm has been initiated (e.g., Johns and Doswell, 1992; Doswell et al., 1996; Fabry, 2006; Púčik et al., 2015; Peters et al., 2017).

Particularly in the domain of nowcasting, the new FCI observations, which will provide detailed and quasi real-time monitoring of boundary layer WV or TCWV and related (convective) cloud formation, could substantially advance the field (e.g.,
Benevides et al., 2015; Van Baelen et al., 2011; Dostalek et al., 2021). On a global, climatological scale, WV is a major contributor to global energy fluxes through the atmosphere and, due to its abundance and absorption of light over a wide range of the solar and terrestrial spectrum, acts as the strongest greenhouse gas(e.g. Trenberth et al., 2003; Schmidt et al., 2010). In the context of climate change, a warmer atmosphere will provide more WV, forming a positive feedback loop, thus further enhancing global warming. Moreover, a moister atmosphere is predicted to produce more severe weather (e.g., Allen and Ingram,
2002; Neelin et al., 2022; Chen and Dai, 2023). Apart from that, WV is considered a disturbing gas for several remote sensing applications, e.g., surface parameter retrievals, for which precise information on WV amounts in the atmosphere is needed for atmospheric correction methods (e.g. Gao et al., 2009; Wiegner and Gasteiger, 2015; Valdés et al., 2021).

The use of these NIR WV absorption measurements for TCWV retrievals is not new. For several decades, the $\rho\sigma\tau$ water vapour absorption region has been researched using radiative transfer models and exploited in TCWV retrieval schemes (e.g.,
Fischer, 1988; Gao and J., 1992; Bennartz and Fischer, 2001; Albert et al., 2005; Lindstrot et al., 2012; Diedrich et al., 2015; Preusker et al., 2021). The focus first lay on ground-based radiometers and soon shifted to airborne and space-borne imagers. The first satellites which carried instruments with dedicated NIR water vapour (WV) bands were almost exclusively on satellite platforms with sun-synchronous, polar orbits and could deliver global daily coverage at a km to hm resolution on a daily basis. Even at a km resolution, NIR TCWV is able to resolve convective phenomena such as horizontal convective rolls or gravity
waves (Carbajal Henken et al., 2015; Lyapustin et al., 2014). The Satellite Application Facility in Support to Nowcasting and Very Short Range Forecasting (NWCSAF) is an organisation funded by EUMETSAT and aims to support meteorological services with satellite data critical for the prediction of high-impact weather (e.g., storms, fog). They commission, develop and maintain software which utilises many weather satellite instruments, including MTG-FCI/Meteosat-12 (García-Pereda et al., 2019).

A NIR TCWV product in the portfolio of NWCSAF's software will greatly benefit the nowcsating and meteorological community at large. Current NIR TCWV products, only available on polar orbits, provide accurate highly-resolved TCWV but lack temporal resolution and may introduce observation biases in climatologies (Diedrich et al., 2016; Carbajal Henken et al., 2020). In contrast, networks which use Global Navigation Satellite Systems (GNSS) to retrieve TCWV have good temporal coverage but lack areal context and cannot provide the coverage of a passive imager.

In this work, we present our TCWV retrieval framework utilizing the novel NIR measurements obtained from MTG-FCI. Our approach builds on established TCWV retrieval frameworks successfully applied to other passive imagers sharing similar spectral band configurations. The differential absorption technique, using the ratio of measurements in the $\rho\sigma\tau$-absorption band and a nearby window band, was previously employed to measurements of the Medium Resolution Imaging Spectrometer (MERIS) onboard Envisat (Bennartz and Fischer, 2001; Lindstrot et al., 2012).



To fill the temporal gap between observations from MERIS and its follow-up instrument Ocean and Land Colour Instrument (OLCI) on-board the Sentinel-3 satellites consistently, the optimal estimation (OE) procedure for MERIS TCWV retrievals was adapted for NIR measurements of the Moderate Resolution Imaging Spectroradiometer (MODIS) on-board Aqua and Terra satellites (Diedrich et al., 2015). More recently, with the launch of the Copernicus Sentinel-3A and Sentinel-3B satellites (Donlon et al., 2012) and onboard OLCI instruments, the retrieval framework has been extended to fully exploit OLCI's

extended spectral capabilities by using multiple bands sensitive to WV absorption (Preusker et al., 2021). Due to the unique technical characteristics of MTG-FCI and the absence of a lengthy and well-calibrated FCI data record at the time of completion of this work, new strategies are imperative for our methodology and its assessment. Among those are an surface reflectance approximation method for the absorption band and the use of OLCI-SLSTR data to mimic FCI data for assessment. In order to test our algorithm with OLCI, the same set of simulations we do for FCI are done for OLCI band 17 (0.865 $\mu$m) and band

19 (0.9 $\mu$m). The same inversion technique is used for both FCI and OLCI.

The structure of this paper is as follows. Section 2 introduces the MTG-FCI data, OLCI-SLSTR data, auxiliary data, and the TCWV reference datasets, along with the associated matchup method. Section 3 details the MTG-FCI TCWV retrieval framework, covering the physical background, forward model, inversion method, and albedo approximation method integral to the algorithm. Section 4 presents the results of the matchup assessments conducted on both local and global scales, along

with initial analyses using a preliminary calibrated MTG-FCI dataset and a representative case study. Section 5 provides a discussion and outlook, and Sect. 6 concludes the paper.

## 2   Data

### 2.1   MTG-FCI Data

Meteosat Third Generation is an operational EUMETSAT satellite mission. It currently consists of one satellite which is in

a geosynchronous orbit at a longitude of 0°. It carries the Flexible Combined Imager (FCI) and the Lightning Imager (LI). FCI is a multispectral instrument which scans with a fast east-west and a slow north-south motion. It has 16 bands which range from the VIS (0.44 $\mu$m) to the thermal infrared (TIR) (13.3 $\mu$m). The full-disk scan service covers a circular area of approximately one-fourth of the Earth's surface within 10 minutes, covering Europe, Africa and parts of the Atlantic and Indian ocean (Durand et al., 2015; Holmlund et al., 2021). In the future, a second MTG-FCI will provide a rapid scan service, which

provides coverage of the upper third of the full-disk within 2.5 minutes, only covering parts of Europe and the Mediterranean. The spatial resolution at sub-satellite point (SSP) of one VIS band at 0.64 $\mu$m and one SWIR band at 2.25 $\mu$m is 0.5 km. The spatial resolution of the other VIS to SWIR bands and the TIR bands at 3.8 $\mu$m and 10.5 $\mu$m is 1.0 km at SSP. The remaining TIR bands have a SSP resolution of 2.0 km. Due to the curvature of the Earth and the coverage of FCI, the real spatial resolution at ground level is slightly lower. For example, the 1 km SSP resolution (VIS, NIR and 10.5 $\mu$m) in Northern Europe is closer

to 2.0 to 3.0 km.

MTG1 has been launched successfully into orbit on 13th of December 2022 and currently the mission is still in the commissioning phase. Because of that we use the latest release of preliminary MTG-FCI level 1c data provided by EUMETSAT



in February 2024 (EUMETSAT, 2024b). They consist of one full-disk scene on 13th January 2024 measured between 11:50
and 12:00 AM UTCZ. They were downloaded from EUMETSAT's sftp server at https://user.eumetsat.int/news-events/news
/new-mtg-fci-test-dataset-mtgtd-505 and more detail on this dataset can be found in EUMETSAT (2024a). At the time of
publication, no cloud mask is readily available for the FCI test data. Therefore, we build a simple cloud mask algorithm. The
majority of the cloud mask algorithm is based on the work presented in Hünerbein et al. (2023). In this publication the authors
adapted and extended cloud masking and typing algorithms developed for NASA's Aqua/Terra Moderate Imaging Spectrom-
eter (MODIS) (Ackerman et al., 2002) to ESA's Cloud Aerosol and Radiation Explorer Mission (EarthCARE) Multi Spectral
Imager (MSI). We adapted a subset of their tests to the FCI bands and estimated new coefficients and thresholds. Eventually, the
cloud mask consists of two tests: a reflectance test and a reflectance ratio or Global Environmental Monitoring Index (GEMI)
test Pinty and Verstraete (1992). For the reflectances, ratios and test the bands VIS_06 (0.640 $\mu$m), VIS_08 and NIR_13 were
used. Reflectances were calculated following Eq. 13.

## 2.2 S3-OLCI/SLSTR Data

Sentinel-3 is an operational COPERNICUS satellite mission of the European Commission, managed by EUMETSAT. It con-
sists of two sister satellites (Sentinel-3A: S3A; Sentinel-3B: S3B) which orbit the Earth at an altitude of 814.5 km, an inclination
of 98.65 ° and a local equator crossing time of 10:00 AM. S3B is phase-shifted to S3A by 140 °. This way the imaging instru-
ments onboard the two satellites achieve global coverage almost daily. The payloads consist of the SAR Altimeter (SRAL),
supported by the Microwave Radiometer (MWR), an visible/near-infrared imager Ocean and Land Colour Instrument (OLCI)
and an visible to thermal infrared imager Sea and Land Surface Temperature Radiometer (SLSTR).

OLCI is a push-broom multispectral imaging spectrometer which consists of five cameras. It measures at 21 discrete bands
ranging for the the VIS (0.4 $\mu$m) to the NIR (1.02 $\mu$m). The swath-width of OLCI is 1215 km at a full SSP resolution of 0.3
km per pixel which is referred to as "Full Resolution". In the "Reduced Resolution", 4 by 4 pixel are aggregated into. 1.2 km
pixels. This is the resolution used in this study.

One OLCI instrument achieves global coverage in three days. In its current two satellite configuration almost complete
global coverage is achieved within one day, with small gaps at the equator. A characteristic of OLCI is an across-track spectral
shift due to the five discrete cameras. This can be corrected for by taking into account the actual central wavelength at each of
the across-track pixels (Preusker et al., in prep.).

SLSTR is a conical scanning multispectral, multiangle radiometer. It measures at eight discrete bands ranging for the mid
VIS (0.55 $\mu$m) to the TIR (12.02 $\mu$m). The nadir-viewing swath width is 1400 km, the oblique/rear-viewing swath width is
740 km. In the VIS and SWIR bands (until 2.25 $\mu$m) SLSTR has a spatial resolution of 0.5 km at SSP, in the TIR (3 $\mu$m to
12.02 $\mu$m) the spatial resolution is 1 km at SSP. Similar to OLCI, SLSTR provides global coverage within two to three days
in nadir view. In oblique view this is the case within three to four days. In the current two satellite constellation, nadir-view
global coverage is achieved within one day.

155 In order to mimic the capabilities of FCI at a similar spatial resolution and with similar spectral characteristics, we merged
some SLSTR observations to the OLCI grid using nearest-neighbour sampling. The SLSTR bands used were bands S5 (1.612



$\mu$m, 0.5 km) and S6 (2.25 $\mu$m, 0.5 km). They have been mapped to OLCI's reduced resolution at 1.2 km. This way, we can test how our algorithm performs globally on real data, before MTG-FCI data at nominal calibration become available publicly. The cloud-mask used to filter out cloudy pixels is the IdePIX cloudmask (Iannone et al., 2017; Wevers et al., 2021; Skakun et al., 2022).

## 2.3 ECMWF ERA5 Forecast Data

Our TCWV retrieval is based on an inversion technique (Sec. 3) which uses a first guess, as well as a priori and ancillary parameter data. These data may come from a climatology or be set to a global climatological value. However, retrieval performance can be greatly increased and speeded up if data fields are passed to the algorithm which are already somewhat close to the solution. This is why we chose to provide the algorithm with fields taken from Numerical Weather Prediction (NWP) forecast fields. These were acquired from the European Centre for Medium-Range Weather Forecasts (ECMWF) ERA5 forecasts initialised at 6 AM UTC and 6 PM UTC of each day (Hersbach et al., 2020). These are different from ECMWF's operational forecasts since they use more assimilated data in the initialisation time step. The forecasts are at a resolution of 0.25° and in 3 h steps. The data fields are interpolated to the observation time and FCI coordinates.

The variables needed are: horizontal wind speed (WSP) calculated from u- and v-component of the horizontal wind speed at 10 m above ground (U10, V10), total column water vapour (TCWV), surface air temperature at 2 m above ground (T2m) and surface air pressure (SP). The data were accessed via the Copernicus Climate Change (C3S) data store (Service and Store, 2023). For testing and algorithm development we used the ERA5 forecasts. In the later operational processing for the NWCSAF GEO software package the operational ECMWF forecasts at a resolution of 0.5° and 1 h steps will be used.

## 2.4 Aerosol Optical Thickness Climatology

One key parameter for the retrieval of TCWV over water is the aerosol optical thickness (AOT). As a first guess for AOT, we use a climatology at a 1° spatial resolution. It was built from monthly means of the Oxford-RAL Aerosol and Cloud (ORAC, Thomas et al. (2009)) aerosol optical properties data retrieved with Sentinel-3 Sea and Land Surface Temperature Radiometer (SLSTR) and Enviromental Satellite (ENVISAT) Advanced Along Track Scanning Radiometer (AATSR) between 2002 and 2022. These data were also accessed via the C3S data store (Copernicus Climate Change Service and Climate Data Store, 2019).

## 2.5 Reference Datasets and Match-up Analysis

In order to verify the credibility of retrieved TCWV we need reference data within the field of view of FCI. There are four established sources of TCWV estimates: radiosondes, ground based GNSS meteorology, microwave radiometers and direct sun-photometry. TCWV from NWP Reanalyses may also be used, but their coarse resolution cannot resolve the fine variabilities found in the WV field at the satellite-pixel scale. Reanalyses may be used to assess the stability of the dataset later on. Unfortunately, until the completion of this work, no MTG-FCI data were available in the final calibration. Because of this,





we processed the spectrally representative test data discussed above. Nevertheless, the performance of our algorithm as well as the accuracy of our calculations require testing on real data. Hence, we processed a 7-year matchup dataset of OLCI observations and Atmospheric Radiation Measurement (ARM, Sisterson et al. (2016)) TCWV over their Southern Great Planes site. Additionally, a month of global OLCI-SLSTR observations was processed with our algorithm, also employing the surface reflectance approximation from Sec. 3.4. These were compared against TCWV data retrieved at sites of the Aerosol Robotic Network (AERONET, Holben et al. (1998)), the Atmospheric Radiation Measurement network (ARM, Turner et al. (2007); Cadeddu et al. (2013)) and the SUOMINET (Ware et al., 2000).

Prior to the analysis, OLCI swaths and ground-based network sites were collocated within 1 km and 30 minutes of a satellite overpass. A square of 11 by 11 pixels around the collocated center pixel were taken into account. These pixels then were screened for convergence of the algorithm, a cost-function below 1 and cloud-cover using the Idepix cloud mask with a buffer of 3 pixels (3.6 km) minimizing the effect of cloud contamination on the TCWV quality assessment. Match-up cases with less than 95% valid pixels were rejected, the central 3 by 3 pixels had to be completely cloud-free.

Both in the assessment of assumptions and the assessment of TCWV quality we use metrics. Their abbreviations are following: N is the number of matchups, MADP is the mean absolute percentage deviation, RMSD is the root mean square deviation, cRMSD is the centered RMSD (i.e., bias corrected), R is the correlation, ODR $\alpha$ and $\beta$ are the orthogonal distance regression coefficients for the intersect and slope, respectively.

## 3 Algorithm Description

### 3.1 Physical Background

The $\rho\sigma\tau$ WV absorption bands are due to the vibrational reaction in a gaseous water molecule being hit by a photon within a specific wavelength range (roughly between 0.9 and 1.0 $\mu$m), see Fig. 1. FCI features a "window" band with a nominal center wavelength of 0.865 $\mu$m (band 4, "VIS_08") and an "absorption" band with a nominal center wavelength of 0.914 $\mu$m (band 5, "VIS_09"). The spectral response functions (SRF) are also shown in Fig. 1.

The absorption of WV in these spectral regions is weak compared to the infrared region at, e.g., 6.7 or 7.3 $\mu$m (traditionally referred to as WV bands). Because of that, a beam of sun light in the $\rho\sigma\tau$ range travels through the atmosphere and is reflected at the Earth's surface and travels trough the atmosphere again and may hit the sensor. This way, the whole column's content of atmospheric WV can be probed. While the signal within the absorption band decreases with WV content, an adjacent window band will be virtually unaffected by any change in WV amount along the line-of-sight (LOS). However, the situation becomes more complex due to multiple scattering within the atmosphere. This photon scattering may lengthen or shorten the light path and thus affecting the amount of measured absorption.

The overall strongest influence factor on the signal measured at the satellite sensor is the surface reflectance at the channel's wavelength, i.e., how much light is reflected at the surface. This depends on the type of surface covering (e.g., vegetation, sand, snow, etc.) and to some degree on the sun and viewing angles. For land cases the spectral albedo in the NIR is well above 0.3 and thus provides a strong signal relative to the absorption by WV. Over the majority of water surfaces, however, the





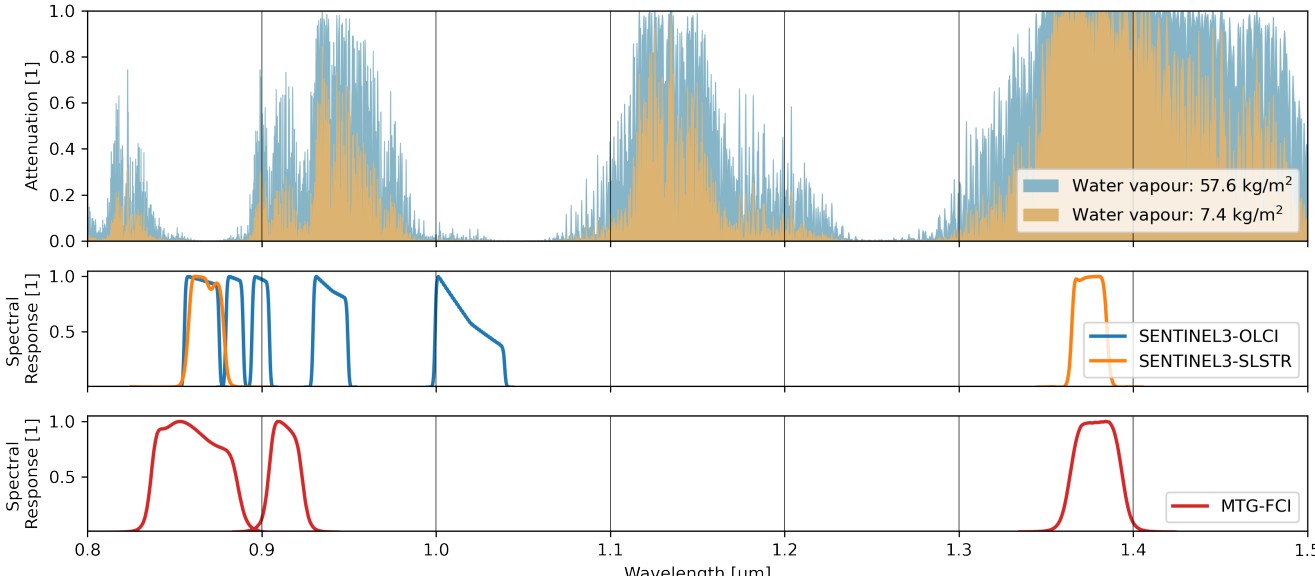

**Figure 1.** Upper panel: The WV attenuation spectrum for an atmosphere with a low TCWV amount in orange (7.4 kg/m$^2$) and high TCWV amount in blue (57.6 kg/m$^2$) with data obtained from the Correlated K-Distribution Model Intercomparison Project (CKDMIP, Hogan and Matricardi (2020)). Center panel: The SRFs in the NIR part of the spectrum for the satellite instruments OLCI (blue) and SLSTR (orange). Lower panel: the SRFs in the NIR for FCI (green).

surface reflectance is often well below 0.03. There is no direct way to measure this spectral albedo, hence an approximation is necessary. This approximation is described in more detail in Sec. 3.4.

A slightly less important effect comes from scattering aerosol layers below a certain level of aerosol optical thickness (AOT). In that case, the effective LOS is shortened by the higher aerosol layer and since the humidity content on average is much lower 225 in the higher troposphere, the absorption is decreased substantially. Over bright surfaces this effect is much less important than over dark surfaces (Lindstrot et al., 2012). Since most natural surfaces over land are bright at 0.9 $\mu$m, the amount of reflected light is high (albedos between 0.2 to 0.9) compared to the atmospheric reflectance. Then, the shielding effect of an average aerosol layer is small.

Under most circumstances, this assumption is not valid for water surfaces, though. On average, the reflectance of a water 230 body in the NIR is well below 0.03. Thus, already slightly scattering layers of aerosol in the troposphere may create the effect described above and lead to an artificially "drier" signal. To a certain degree this effect can be corrected for, by simulating for an aerosol layer with a specific aerosol optical depth in the algorithm. However, for this, the effective height of the aerosol layer needs to be estimated and this is a challenge in and of itself.

Another important aspect over water surfaces is their reflectance's dependency on wind-speed and viewing geometry. High 235 wind speeds (rough surface) create a low reflectance peak (still higher than without sun glint) spread out over a range of



observation geometry angles. Low wind speeds (calm surface) create more of a peak with a much higher reflectance over a small range of an observation cone, similar to a mirror.

In both land and water surface cases the atmospheric temperature and pressure profiles as well as the observation geometry also play a role due to temperature- and pressure-dependent line broadening.

## 3.2 Forward Model

The first step in our framework is to run radiative transfer simulations (RTS) for a set of complete and comprehensive atmospheric, surface and geometric conditions as described in the previous section and summarised in Table 1 and 2.

For the simulation of top-of-atmosphere (TOA) reflectances we used the Matrix Operator Model (MOMo, Fell and Fischer (2001); Hollstein and Fischer (2012)). These simulations are finally sorted into two land look-up-tables (LUT) for land surfaces and water surfaces, respectively.

The following variables and variable names are used within the algorithm and throughout this report: total column water vapour (TCWV), isotropic surface albedo (alb,$\rho_{BOA}(\lambda)$), AOT, horizontal wind speed at 10 m (WSP), surface pressure (SP), air temperature at 2 m (T2M), viewing/satellite zenith angle (SATZ), satellite/viewing azimuth angle (SATA), sun/solar zenith angle (SUNZ), solar/sun azimuth angle (SUNA), and relative azimuth (RAZI). To avoid ambiguity in its definition we use Eq. 1:

$$RAZI = arccos(cos(SUNA) * cos(SATA) + sin(SUNA) * sin(SATA)) \tag{1}$$

The measurements we use to retrieve TCWV is the window channel observation $nL_{TOA}(0.865\mu m)$ (a normalised radiance) and the absorption channel observation $\tau_{TOA}(0.914\mu m)$, a pseudo optical thickness, see Eq. 3. The normalised radiance is calculated following:

$$nL_{TOA}(\lambda) = \frac{L_{TOA}(\lambda)}{F_0(\lambda)} \tag{2}$$

where $F_0$ is the spectral solar irradiance.

The pseudo optical thickness $\tau_{pTOA}$ is calculated following:

$$\tau_{pTOA}(\lambda) = -a - \frac{log(\frac{nL_{TOA}(\lambda)}{nL_{TOA}^*(\lambda)})}{\sqrt{AMF}} \cdot b \tag{3}$$

where $AMF$ is the air mass factor, $a$ and $b$ are the so-called correction coefficients which may correct for a systematic bias discovered in a validation against reference TCWV observations. Preusker et al. (2021) have obtained these coefficients by minimizing the differences between simulated and measured OLCI observations using ARM-SGP.C1-MWR TCWV as an input (see Preusker et al. (2021) for details). $a$ and $b$ for band 19 (at 0.9 $\mu$m) were estimated to be -0.008 and 0.984, respectively from the results shown in Sec. 4.1.

For FCI another MWR TCWV reference (and associated TCWV observations) will be necessary, we intend to use the reference sites such as Meteorological Observatory Lindenberg – Richard Assmann Observatory (MOL–RAO) (Knist et al.,



| Variable Name | Range and units |
|---|---|
| TCWV | 0.1 kg/m$^2$ to 75.0 kg/m$^2$ |
| ALB | 0 to 1 |
| | |
| T2M | standard atmospheric profiles 1 to 5$^*$ |
| SP | 500 hPa to 1050 hPa |
| SUNZ | 0° to 90° |
| SATZ | 0° to 85° |
| RAZI | 0° to 180° |

**Table 1.** Land Surface Setup for MOMo. TCWV = Total Column Water Vapour; alb = Surface albedo; T2M = Air Temperature at 2 m; SP = Surface Pressure at 2 m; SUNZ = Sun Zenith Angle; SATZ = Satellite Zenith Angle; RAZI = Relatie Azimuth Angle. $^*$Standard profiles from Anderson et al. (1986).

| Variable name | Range and units |
|---|---|
| TCWV | 0.1 kg/m$^2$ to 75.0 kg/m$^2$ |
| AOT | 0.1 to 1.1 at 700 to 1000 m height |
| WSP | 2 m/s to 15 m/s |
| T2M | standard atmospheric profiles 1 to 5$^*$ |
| SP | 950 hPa to 1050 hPa |
| SUNZ | 0° to 90° |
| SATZ | 0° to 85° |
| RAZI | 0° to 180° |

**Table 2.** Water Surface Setup for MOMo. AOT = Aerosol Optical Thickness; WSP = Wind Speed. $^*$Standard profiles from Anderson et al. (1986).

2022), the Cabauw Experimental Site for Atmospheric Research (CESAR) (Van Ulden and Wieringa, 1996) or ARM – Eastern North Atlantic (ENA) (Mather and Voyles, 2013).

$nL_{TOA}$ is calculated following Eq. 2 and $nL_{TOA}^*$ refers to the normalised radiance corrected for the influence of water vapour absorption. $nL_{TOA}^*$ needs to be approximated using other available information (e.g., a climatology atlas, neighbouring window channels). Here, we use a more elaborate technique, described in Subsection 3.4. The $AMF$ is calculated, following:

$$amf = \frac{1}{cos(SUNZ)} + \frac{1}{cos(SATZ)} \tag{4}$$

The aerosol mixtures and their optical properties have been calculated using the OPAC software package (Optical Properties of Aerosols and Clouds, Hess et al. (1998)). Within their documentation you can find details on the used aerosol mixtures for the two types chosen for the land and water surface simulations. Over land we used the aerosol mixture "continental average", over ocean we used the aerosol mixture "maritime clean". The standard atmospheric profiles used to set the vertical distribution of temperature and moisture were taken and adapted from Anderson et al. (1986). The numbers refer to: 1. mid-latitude summer, 2. mid-latitude winter, 3. sub-Arctic summer, 4. sub-Arctic winter, 5. tropical. The configurations used for the simulations are listed in Tables 1 and 2.

This set of simulations is then sorted into a multi-dimensional look-up table (LUT). This LUT can then be used to simulate an observation ($\boldsymbol{y}$) for a given set of states ($\boldsymbol{x}$) and parameters ($\boldsymbol{p}$) using an interpolator. This is referred to as the forward model $\boldsymbol{F}$. With this forward model, we can estimate an sensor's observation for a given set of states, following:

$$\boldsymbol{y} = \boldsymbol{F}(\boldsymbol{x}, \boldsymbol{p}) + \epsilon \tag{5}$$

$\epsilon$ denotes the measurement and forward model error.



### 3.3 Inversion Using Optimal Estimation

Eq. 5 can be inverted in order to retrieve a state describing an observation from a measurement by a sensor. There are various ways of performing an inversion. We chose to follow the optimal estimation (OE) approach for atmospheric inverse problems described by Rodgers (2000). In essence, this inversion is based on the principle of minimizing the cost function $\boldsymbol{J}$ by iteratively changing the initial first guess of a state or the state of the prior iteration step. $\boldsymbol{J}$ is calculated following:

$$\boldsymbol{J}(\boldsymbol{x}) = \frac{1}{2}(\boldsymbol{y} - \boldsymbol{F}(\boldsymbol{x}))^T \boldsymbol{S_e}^{-1}(\boldsymbol{y} - \boldsymbol{F}(\boldsymbol{x})) + \frac{1}{2}(\boldsymbol{x_a} - \boldsymbol{x})^T \boldsymbol{S_a}^{-1}(\boldsymbol{x_a} - \boldsymbol{x}) \tag{6}$$

where $\boldsymbol{x_a}$ is the a priori state, $\boldsymbol{S_a}$ is the a priori error covariance matrix associated with $\boldsymbol{x_a}$ and $\boldsymbol{S_\epsilon}$ is the measurement error covariance matrix.

We assume the forward model and a priori knowledge to be bias-free and the inverse problem within the confines of the assumptions stated in previous Sections 3.1 and 3.2 to be near-linear. Then, we can use the Gauss-Newton approach to calculate the next step of the iteration $\boldsymbol{x_{i+1}}$ following:

$$\boldsymbol{x_{i+1}} = \boldsymbol{x_i} + (\boldsymbol{S_a}^{-1} + \boldsymbol{K_i}^T \boldsymbol{S_\epsilon}^{-1} \boldsymbol{K_i})^{-1}[\boldsymbol{K_i}^T \boldsymbol{S_\epsilon}^{-1}(\boldsymbol{y} - \boldsymbol{F}(\boldsymbol{x_i})) - \boldsymbol{S_a}^{-1}(\boldsymbol{x_i} - \boldsymbol{x_a})] \tag{7}$$

where $\boldsymbol{K}$ is the Jacobian which contains the partial derivatives of each measurement to each state at step $i$ (i.e., $\boldsymbol{K_i} = \partial \boldsymbol{F}(\boldsymbol{x_i})/\partial \boldsymbol{x_i}$). The iterative process is stopped if either the maximum number of allowed steps is reached or the following criterion is met by the retrieved state $\boldsymbol{x_{i+1}}$:

$$(\boldsymbol{x_i} - \boldsymbol{x_{i+1}})^T \hat{\boldsymbol{S}}_i^{-1}(\boldsymbol{x_i} - \boldsymbol{x_{i+1}}) \leq n \cdot \epsilon \tag{8}$$

where $\hat{\boldsymbol{S}}$ is the retrieval error-covariance. In this case we retrieve the true state $\boldsymbol{x}$. Under the assumption that the uncertainties of state, forward model and measurement are distributed Gaussian, we may calculate the associated uncertainty $\hat{\boldsymbol{S}}$ of $\boldsymbol{x}$:

$$\hat{\boldsymbol{S}} = (\boldsymbol{S_a}^{-1} + \boldsymbol{K_i}^T \cdot \boldsymbol{S_\epsilon}^{-1} \cdot \boldsymbol{K_i})^{-1} \tag{9}$$

More details on the process of optimal estimation within a TCWV retrieval framework can be found in Preusker et al. (2021) and El Kassar et al. (2021). Factors such as the uncertainty of the prior knowledge or the measurement (e.g., the signal-noise-305 ratio; SNR) are taken into account. Such uncertainties may either be set to values which correspond to the actual covariances within a given variable. However, the uncertainties may also be used as tuning parameters in order to make the algorithm lean more towards the measurement or more towards the prior knowledge. Over land surfaces, we set the apriori-uncertainty of the TCWV very high (16 kg/m$^2$) since the information content of the band is high over bright surfaces. Over ocean, the apriori-uncertainty was set much lower (2.5 kg/m$^2$). Here the information content is much lower, due to the strong absorbance 310 of the water body at the band wavelengths.

The retrieval procedure is as follows. First, FCI data are read in and then the necessary auxiliary fields (e.g., first guess TCWV, T2M, WSP) are read and interpolated to satellite resolution. As input for the prior knowledge and first guess we use the forecast at 6 AM or 6 PM of the ECMWF ERA5 at a resolution of 0.25°. As a first guess for AOT we use the ORAC





climatology at a $1°$ resolution. In the next step, the measurements (e.g. reflectances, $\tau_{pTOA}$, etc.) and the cloud mask are
calculated. Then, a land and water processing mask is made. Here, pixels which are considered cloudy or which are closer to
the night than the day, i.e., where the solar elevation angle is too slant ($SUNZ > 80°$), are filtered out.

Subsequently, the interpolation is applied to the LUT to yield the simulated FCI measurements for the set of atmospheric,
geometric and state variables for each pixel. These are compared against the real FCI measurements. Iteratively, the state is
adjusted and again fed into the LUT-interpolation in order to move closer to the measurement within a pre-defined uncertainty
and convergence criterion. Once this is reached, this state is passed out of the algorithm and these pixels are marked as
converged. If a specific number of iterations is exceeded (8 over land, 6 over ocean), the algorithm stops and these pixels are
marked as not-converged. Furthermore, an estimate of the associated uncertainty in the retrieved state is provided as part of the
algorithm's output.

After the processing has finished for all pixels, data are only marked as valid if their cost is below a threshold (currently $< 1$
for land, $< 1.5$ for ocean) and if the convergence criterion has been met. This way, some cloudy pixels which have been missed
by the cloud mask or pixels which contain a scattering aerosol layer may be filtered out. This is due to the fact that NIR-TCWV
retrievals in the presence of elevated cloud or aerosol layers often lead to a substantial underestimation of TCWV, compared to
the TCWV found in the prior/first guess. This leads to the cost function to become extremely high.

### 3.4  Estimation of $nL_{TOA}{}^{*}$ with Principle Component Regression

The spectral albedo is the ratio of outgoing irradiance against incoming irradiance at one specific wavelength at a surface.
Over clear, open ocean surfaces, the spectral albedos between the window and absorption bands are relatively close to each
other (i.e., $nL_{BOA}(0.865) \approx nL_{BOA}(0.914)$). Over most other surfaces however, this is not the case. The assumption, that
the ratio of $nL_{TOA}(0.914\mu m)$ and $nL_{TOA}(0.865\mu m)$ would yield a reliable estimate of the water vapour, would thus lead to
substantial over and under-estimations, depending on the spectral slope.

In order to calculate the pseudo optical depth at the absorption band we need an accurate estimate of the spectral slope. For
satellite sensors such as MODIS or OLCI, the WV absorption bands have at least two accompanying window bands (i.e. at
0.865, 0.885, 1.02 or 1.2 $\mu$m). FCI does not have such close additional window bands. Hence, another technique to estimate
the spectral slope is needed. The approach we chose to achieve this is based on the reconstruction of spectra from a principal
component regression (PCR). This approach is already used with reasonable success in the estimation of BRDF and reflectance
spectra within RTTOV (Vidot and Borbás (2014)). Their approach was used as a blue print for our spectral slope estimation.

A spectrally high-resolved spectrum can be reconstructed from much fewer nodes with a so-called PCR. In order to do
this, a collection of high resolved reflectance spectra with a high representativity is needed. The ECOSTRESS spectral library
version 1.0 provided by the United States of America Geological Service (USGS) is such a collection of spectral reflectances
for individual materials and/or mixtures at a high spectral resolution (Meerdink et al., 2019). The library consists of spectra
for the following material groups: human-made, rock, soil, mineral, photosynthetic vegetation, non-photosynthetic vegetation,
water (which includes fresh-water, ice and snow). A small selection of these spectra is depicted in the upper part of Fig. 2,
below, the SRFs of a selection of sensors are shown.





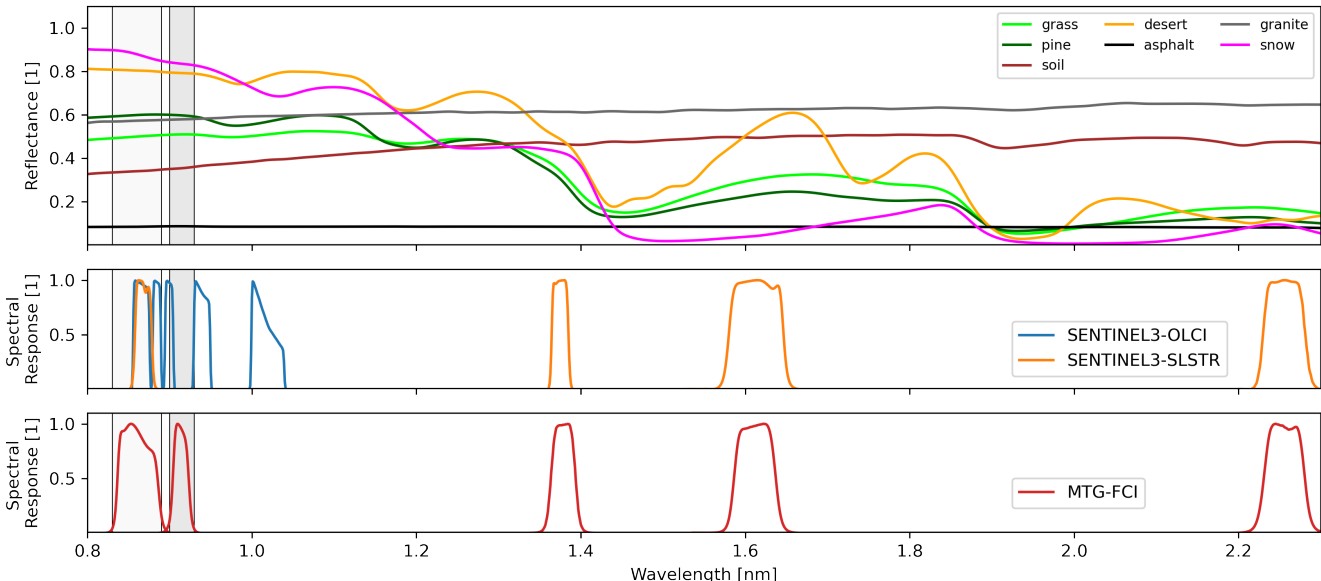

**Figure 2.** Upper panel: overview of a selection of surface reflectance spectra from Meerdink et al. (2019), the labels are representative and not the actual spectra designations. Central panel: the SRFs of OLCI (blue) and SLSTR (orange). Lower panel: the SRFs of FCI (red) and METImage (green).

Only spectra which were available between 0.4 and 2.35 $\mu$m were taken into account and linearly interpolated to a spectral resolution of 0.001 $\mu$m. Furthermore, we used sub-samples of each category, rather than the full collection to avoid a sampling
bias towards rock, mineral and vegetation spectra. From this collection, the principle components (the eigenvectors, PCs) are calculated and sorted by their associated eigenwert.

Instead of using our PCs to reconstruct a spectrally high resolved spectrum we just reconstruct the reflectance of two bands: the NIR window band at 0.865 $\mu$m and the NIR absorption band at 0.914 $\mu$m referred to as the *target*. Following the nomenclature of Vidot and Borbás (2014), $\boldsymbol{R}_{target}$ is the vector of reflectance spectra folded to the target SRFs, $\boldsymbol{c}_{win}$ is the regression
coefficient vector (also referred to as weigths) from the window SRFs and $\boldsymbol{U}_{target}$ is the matrix of the selected PCs of the high-resolution reflectance spectra, folded to the target SRFs.

$$\boldsymbol{R}_{target} = \boldsymbol{c}_{win}\boldsymbol{U}_{target} \tag{10}$$

Using the Moore-Penrose Pseudo inverse, the regression coefficient $\boldsymbol{c}_{win}$ follows:

$$\boldsymbol{c}_{win} = \boldsymbol{R}_{win}\boldsymbol{U}_{win}^{T}(\boldsymbol{U}_{win}\boldsymbol{U}_{win}^{T})^{-1} \tag{11}$$

An optimal configuration of number of PCs and bands was then found by comparing different band combinations with several numbers of PCs. For this we reconstructed all available spectra (which were used in the PCR) at the target bands from





the folded spectra at the window bands. Using this approach, the optimal configuration for MTG-FCI was found with the use of five *window* bands (i.e., no WV attenuation) in the VIS to SWIR (0.51, 0.64 ,0.865, 1.61, 2.25 $\mu$m) and only the first four PCs. We are able to reproduce the "real" surface reflectance at the absorption and window band with a bias of 0.0045 and 0.0038 and a RMSD of 0.016 and 0.02, respectively. By folding the PCs to the SRFs of another instruments makes this matrix applicable to other sensor with similar bands, as shown in Fig. 2.

From the reconstructed surface reflectances we calculate the slope $r$:

$$r = \frac{\rho(0.914)}{\rho(0.865)} \tag{12}$$

This ratio is then multiplied with the $nL_{TOA}(0.865\mu m)$ at $0.865\mu$m in order to yield a more accurate estimate of $nL_{TOA}^{*}$ at the absorption band.

In order to estimate the spectral slope in the PCR, the normalised radiances at the window channels need to be transformed into reflectances:

$$\rho_{TOA}(\lambda) = \frac{nL_{TOA}(\lambda) \cdot \pi}{cos(SUNZ)} \tag{13}$$

The underlying assumption is that within a window band it is valid that $\rho_{BOA}(\lambda) \approx \rho_{TOA}(\lambda)$ and the assumption of isotropic reflectance. Given a sufficiently bright surface and outside of the influence of thick, scattering layers (e.g., clouds, aerosols) or outside of very slant viewing geometries ($SATZ > 82°$), this is the case. Over water surfaces, the influence of scattering processes in the atmosphere is much stronger. Hence, the uncertainties over water pixels are higher. Furthermore the influence of water constituents (e.g. sediment, pigments) on the water reflectance spectrum in the NIR has not been taken into consideration. The PCA training dataset almost exclusively consisted of terrestrial reflectances and only a few fresh water reflectances.




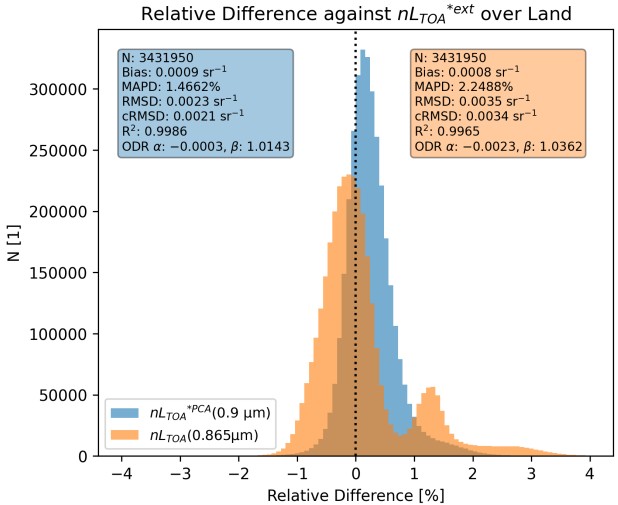
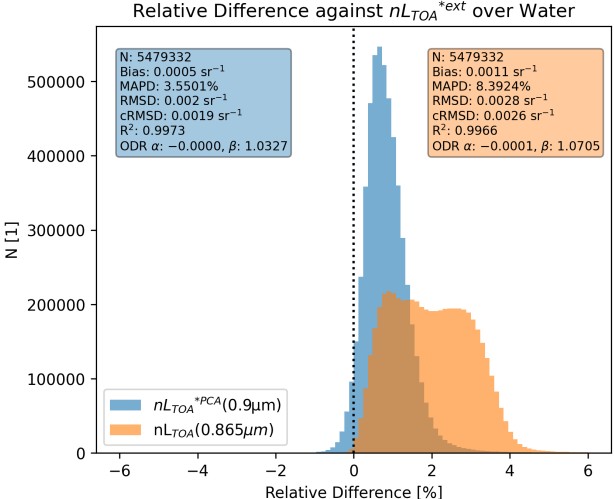

(a) Relative differences over land surfaces for a random subset of pixels of all OLCI/SLSTR data in June 2021. In blue, the relative difference between $nL_{TOA}{}^{*PCR}$ and $nL_{TOA}{}^{*ext}$ and in orange the relative difference between $nL_{TOA}(0.865\mu m)$ and $nL_{TOA}{}^{*ext}$.

(b) Relative differences over ocean surfaces for a random subset of pixels of all OLCI/SLSTR data in June 2021. In blue, the relative difference between $nL_{TOA}{}^{*PCR}$ and $nL_{TOA}{}^{*ext}$ and in orange the relative difference between $nL_{TOA}(0.865\mu m)$ and $nL_{TOA}{}^{*ext}$.

**Figure 3.** Relative differences between two proposed $nL_{TOA}{}^{*}$ against the extrapolated $nL_{TOA}{}^{*ext}$ as used in the COWa algorithm over land and water surfaces, respectively. The associated metrics in the corresponding colours are found in the top corners. The solid black line indicates 0% relative deviation.

Using OLCI as a test-bed provides us a testing possibility: assessing the performance of our PCR to estimate $\tau_{pTOA}$ in comparison to directly using the window ($nL_{TOA}(0.865\mu m)$). In the Copernicus Sentinel-3 OLCI Water Vapour product (COWa) algorithm, the reference used for estimating $nL_{TOA}{}^{*}(0.9\mu m)$ is extrapolated to the absorption band 0.9 $\mu$m from the two window bands 0.865 and 0.885$\mu$m (Preusker et al., 2021). For the $\tau_{pTOA}(0.94\mu m)$ the absorption band surface reflectance is interpolated from the two window bands 0.885$\mu$m and 1.020$\mu$m. This is substantially closer to the the "real" surface reflectance than using a the PCR. Hence, we compare $nL_{TOA}{}^{*PCA}(0.9\ \mu m)$ against $nL_{TOA}{}^{*ext}(0.9\ \mu m)$ from the extrapolation using the
two adjacent window channels. The comparison shown in Fig. 3a and 3b reveals that the vast majority of points lie close to 0 % line for both land and water pixels, albeit with a positive bias. In contrast, using the 0.865$\mu$m normalised radiance by itself would yield much worse results. I.e., a strong bi-modal distribution over land and a weaker bi-modal distribution with a wide spread over water (see Fig. 3a and 3b).

On average, there is a small positive bias in $nL_{TOA}{}^{*PCA}(0.9\ \mu m)$ both over land (+0.3%) and water (+0.8%). Over land pixels the 98th percentile of the relative percentage deviation is 1.7 % against the 2.6 % when using $nL_{TOA}(0.865\mu m)$ as $nL_{TOA}{}^{*}$. Over water pixels the 98th percentile of the relative percentage deviation lies at 2.2% whereas this value is 4% when using $nL_{TOA}(0.865\mu m)$ as $nL_{TOA}{}^{*}$. On average, an increase of 1% in $nL_{TOA}{}^{*}(0.9\mu m)$ roughly translates to a 1.6% increase (i.e., 0.9 kg/m$^2$) of TCWV estimate. A correction of this bias may be possible but since such an analysis cannot be carried out



using FCI we decided against it. Because the PCR performed significantly better than the window channel itself, we decided to use $nL_{TOA}{}^{*PCR}(0.9\ \mu m)$ to calculate $\tau_{pTOA}$ over both land and water surfaces. Despite the slight deviations, the PCR approach remains a good technique in order to reduce the impact of the spectral slope as much as possible.

     This can also be demonstrated using a test processing of TCWV from a single day of OLCI/SLSTR observations. Here, we compared the retrievals from using each $nL_{TOA}{}^{*ext}$, $nL_{TOA}{}^{*PCR}$ and $nL_{TOA}(0.865\mu m)$ to calculate $\tau_{pTOA}$ as input to

the algorithm. In order to only see the influence on precision of TCWV, both datasets have been bias-corrected. The results are shown in Fig. 4. Over land surfaces the bi-modal distribution in using $nL_{TOA}(0.865\mu m)$ persists with large spread and systematic over- and under estimations. Over ocean, the difference between the two approaches is even more pronounced. Both MAPD and RMSD indicate that using the PCR to estimated $nL_{TOA}{}^{*}$ instead of simply the window channel for the calculation of $\tau_{pTOA}$ improves the retrieval substantially.

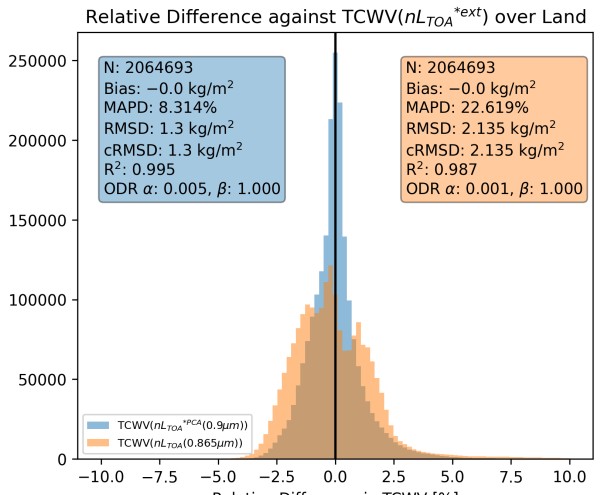

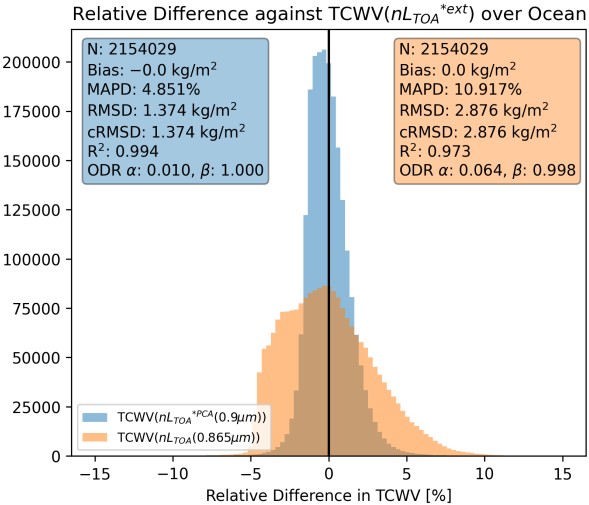

(a) TCWV using $nL_{TOA}{}^{*PCR}$ from the PCR against the extrapolated $nL_{TOA}{}^{*}(0.9\mu m)$ over land pixels for a random subset of data in June 2021.

(b) Relative difference against $nL_{TOA}{}^{*}(0.9\mu m)$ from the PCR against the extrapolated $nL_{TOA}{}^{*}(0.9\mu m)$ over water pixels for a random subset of all data in June 2021.

**Figure 4.** Relative difference between TCWV retrieved with $\tau_{pTOA}$ calculated from extrapolated $nL_{TOA}{}^{*ext}$ and $nL_{TOA}{}^{*PCR}$ from the PCR in blue and $\tau_{pTOA}$ calculated from $nL_{TOA}(0.865\mu m)$ in orange. The TCWV has been bias-corrected against the reference (TCWV using $nL_{TOA}{}^{*ext}$). The data are for a random subset of one day in June 2021. The associated metrics in the corresponding colours are found in the top corners. The solid black line indicates 0% relative deviation.

In very rare cases (<0.1%), there are large deviations (>5%). We assign these cases to 1) geolocation and unphysical spectral matches between OLCI and SLSTR, 2) coastal pixels with a mixed contribution by land and water surfaces, 3) water-constituents changing the NIR reflectance of the water surface substantially or 4) adjacency effects, the brightening effect of dark pixels by diffuse radiation from neighbouring bright pixels and 5) unidentified clouds. Yet, these rare deviations are still lower than the extreme deviations found by using the window band at 0.865 $\mu m$ itself.



 # 4   Results

## 4.1   Sentinel3 OLCI and OLCI/SLSTR data

A first initial test for our forward model as well as the inversion technique was the application to an existing matchup database used for the validation and quality control of COWa. OLCI measurements were spatio-temporally collocated with the ARM network site Southern Great Plains (SGP) positioned in the Mid-West of the United States of America (USA). The dataset is
limited to one location only and runs from 2016 until 2023. Since SLSTR measurements are missing from this dataset, the approximation of $nL_{TOA}{}^{*PCR}$ in the absorption band using the PCA regression could not be done. Instead, we choose the same approach as COWa: extrapolate $nL_{TOA}$ from band 17 (0.865 $\mu$m) and band 18 (0.885 $\mu$m) to band 19 (0.9 $\mu$m).

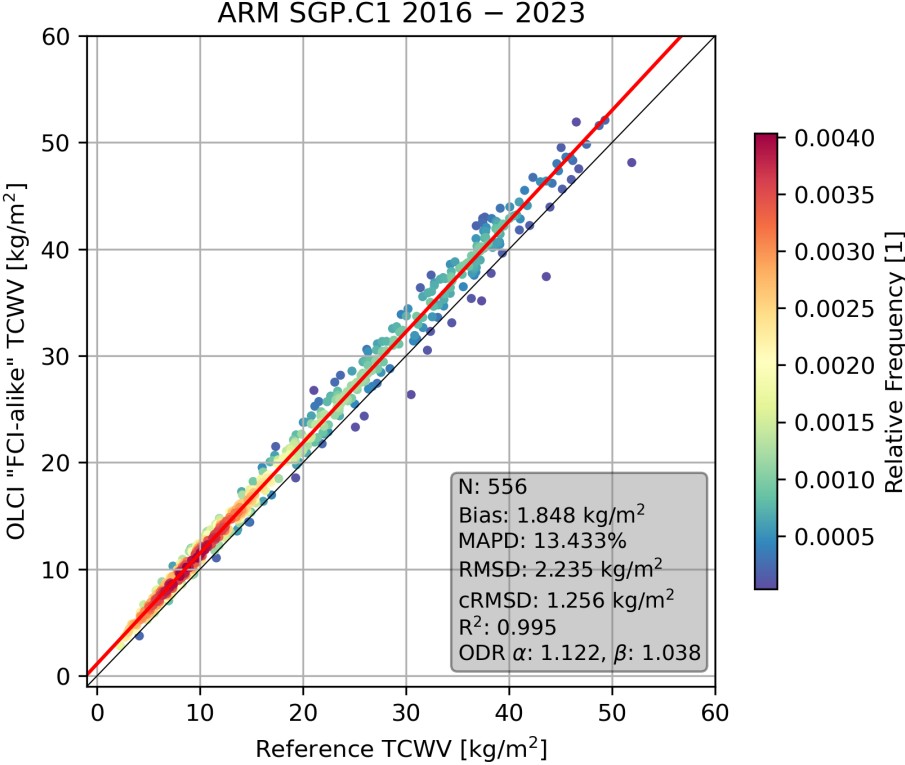

**Figure 5.** Comparison of S3OLCI "FCI-like" TCWV (using $nL_{TOA}{}^{*ext}$) against ARM TCWV at the SGP site, coloured with the relative frequency of occurrence. The solid black line presents the 1:1 line, the red line marks the ODR curve.

The analysis of the two-band TCWV yielded a strong correlation with a Pearson correlation coefficient of 0.995. The bias of 1.848 kg/m$^2$, orthogonal distance regression (ODR) coefficients, i.e., offset ($\alpha$) and slope ($\beta$) of 1.122 and 1.038, respectively,
indicate a slight wet bias. The cRMSD of 1.256 kg/m$^2$, RMSD of 2.235 kg/m$^2$ and MAPD of 13.433% still indicate slight spread.



In a next step, we did a global processing of the OLCI-SLSTR synergy dataset for the month June in 2021. This has been done in order to assess the quality of the two-band approach and the LUT-inversion in combination with the PCR approach to estimate $\tau_{pTOA}(0.9\mu m)$. Fig. 6a shows a global, daily composite from this dataset for 6th of June 2021. As a visual reference, the daily average of the ERA5 reanalysis as well as the COWa TCWV product are shown below. Since COWa was processed with the IdePIX cloudmask using an extra buffer of 2 pixels around each cloudy identified pixel, the grey areas are slightly larger in Fig. 6c. Significant features such as the dry airmasses at the poles and wet airmasses along the Inter-tropical Convergence Zone (ITCZ) exhibit very similar patterns.

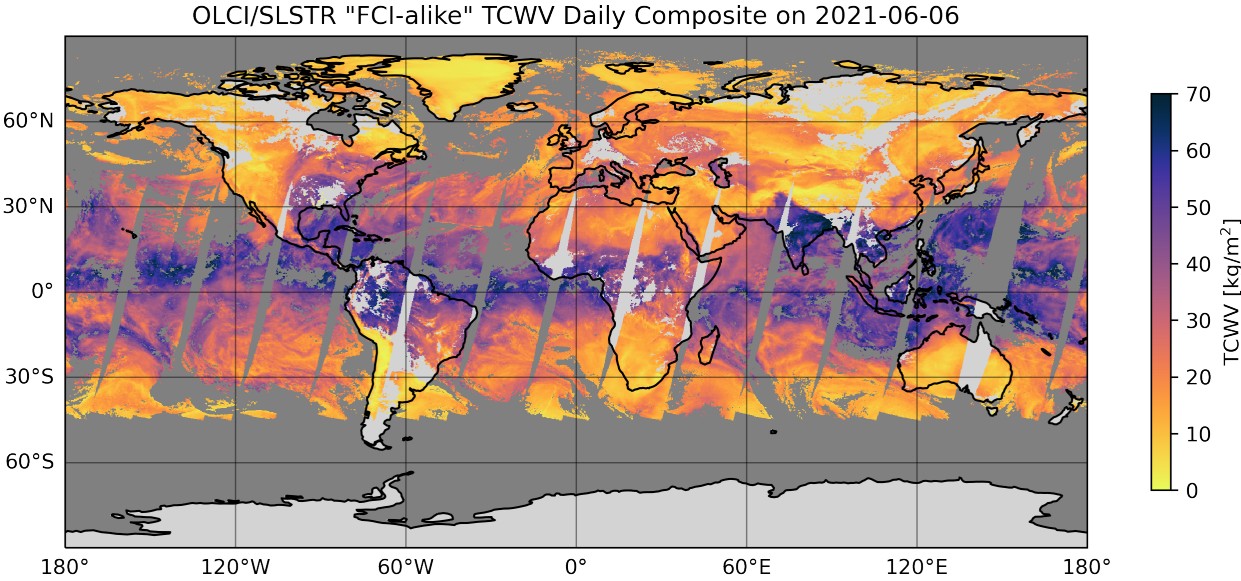

(a) Daily, global 0.05° composite of TCWV processed from S3A and B OLCI/SLSTR data in an "FCI-like" configuration.

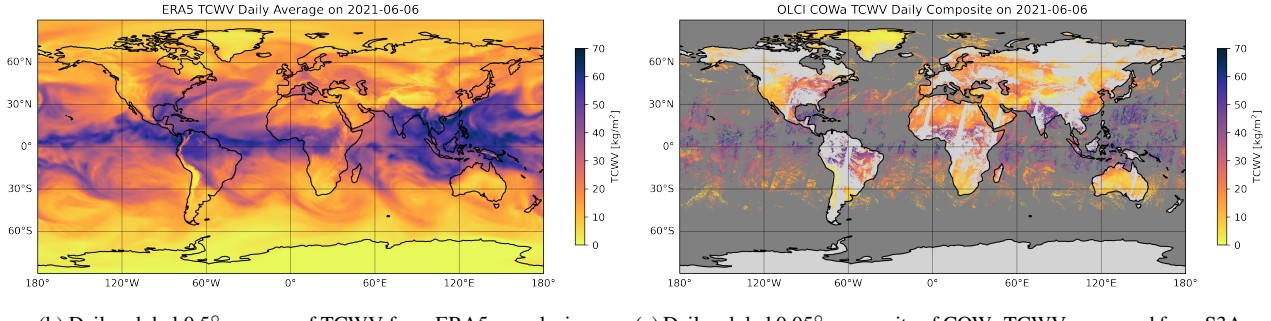

(b) Daily, global 0.5° average of TCWV from ERA5 reanalysis.

(c) Daily, global 0.05° composite of COWa TCWV processed from S3A and B OLCI data.

**Figure 6.** Overview of different global TCWV products for the 6th of June 2021. Grey areas are due to cloud-cover or the retrieval not converging. Dark-grey indicates water surfaces, light-grey indicates land surfaces.



We used this global collection of TCWV to compare against three different reference observations of TCWV as well. For
this matchup analysis we followed the same procedure as for the matchup analysis against the single ARM SGP station. The
results of the comparison are depicted in Fig. 7a to 7d. Fig. 7a shows the positions of the ground-based reference sites with
at least one valid matchup and with their associated network name. For the ARM network only 2 stations were available for
June 2021. Thanks to AERONET and SUOMINET a wide range of different climate zones and atmospheric conditions can
be covered within only one month. The comparison of 281 valid matchups against AERONET in Fig. 7b reveal a wet bias of
3.121 kg/m$^2$, a MAPD of 36.705% a RMSD of 3.906 kg/m$^2$, cRMSD of 2.35 kg/m$^2$, $r^2$ of 0.952 and ODR offset and slope of
1.558 and 1.107 respectively. The analysis results for 11 valid matchups against ARM MWR observations can be seen in Fig.
7c and show a wet bias of 1.23 kg/m$^2$, a MAPD of 6.98 %, a RMSD of 2.55 kg/m$^2$, a cRMSD of 2.24 kg/m$^2$, $r^2$ of 0.97 and
ODR offset and slope of 1.79 and 0.98, respectively. In the comparison of 189 matchups against SUOMNINET GNSS TCWV
we find a wet bias of 2.006 kg/m$^2$, a MAPD of 41.243 %, a RMSD of 2.749 kg/m$^2$, a cRMSD of 1.879 kg/m$^2$ a $r^2$ of 0.96 and
ODR offset and slope of 2.194 and 0.983, respectively.



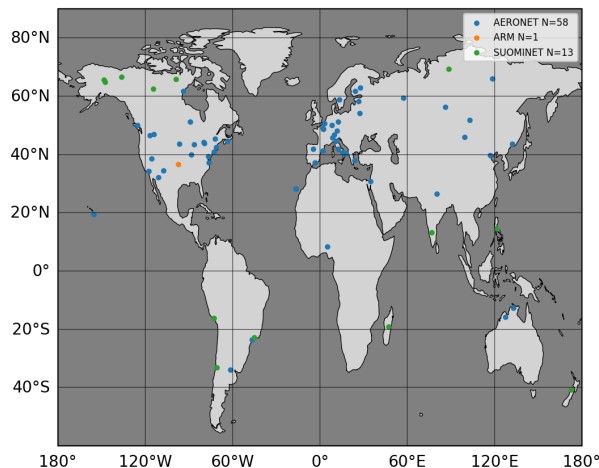

(a) Geographical distribution and their sum of all reference sites from which reference TCWV datasets are used in the matchup analysis.

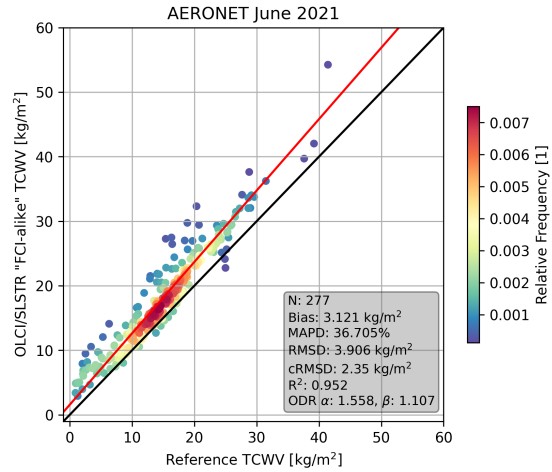

(b) Comparison of OLCI/SLSTR "FCI-alike" TCWV against AERONET TCWV, coloured with the relative frequency of occurrence.

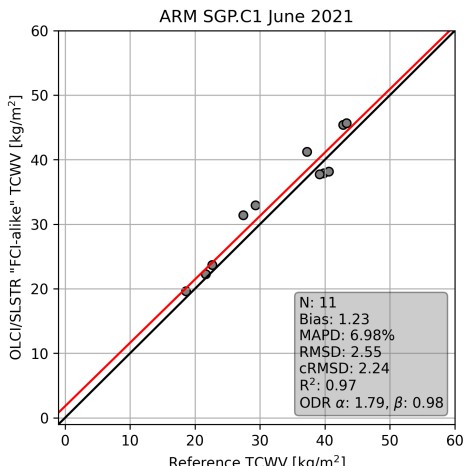

(c) Comparison of OLCI/SLSTR "FCI-alike" TCWV against ARM at SGP.C1 TCWV, coloured with the relative frequency of occurrence.

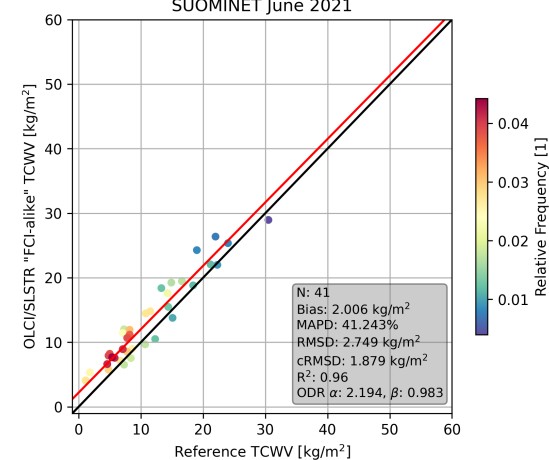

(d) Comparison of OLCI/SLSTR "FCI-alike" TCWV against SUOMINET TCWV, coloured with the relative frequency of occurrence.

**Figure 7.** Matchup Analysis of OLCI/SLSTR "FCI-alike" TCWV against globally distributed reference sites. The solid black lines presents the 1:1 line, the red lines mark the respective ODR curves.

## 4.2 MTG-FCI Data

In order to test our algorithm with regard to future nominal FCI data, we applied the first prototype on test data provided by EUMETSAT. Since this dataset is still preliminary this is neither a definitive nor a quantitative assessment. Rather, it serves to





check the processors performance with real data and check the product for any unexpected behaviour and/or defects. The data
were gathered on the 13th of January 2024 at 11:50 AM UTC. The full-disk natural colour RGB and TCWV product processed
are depicted in Fig. 8a and 8b, respectively.

(a) FCI red-green-blue composite.

(b) Retrieved FCI TCWV field.

(c) Relative difference between FCI TCWV and ERA5 Reanalysis TCWV.

(d) Averaging Kernel corresponding to the retrieved TCWV.

**Figure 8.** Fulldisk visualisation of TCWV and related products, processed from FCI data aquired on 13 January 2024. Dark grey marks land
surfaces, light grey marks water surfaces.

Areas identified as cloudy or which have too low sun or viewing zenith angles have not been processed. Pixels where
the maximum number of iterations were exceeded, or the convergence criterion was not met or the cost function exceeded
a threshold (0.5 over land, 1.0 over water) were masked out. Since the cloud-mask algorithm was set very loose, there are





some cloud-contaminated pixels which have been processed with the algorithm. In parallel processing, the running time of one full-disk scene with on a machine with 64 GBs of RAM and a 12 core CPU lies below 5 minutes. In single processing, the running time of a single chunk takes about 30 to 50 seconds. This includes the cloud-masking, PCR and inversion.

     Similar to Fig. 6, we can see lower TCWV values at the higher latitudes, and much higher values all around the ITCZ in Fig. 8b. Since we are in January, the ITCZ is positioned further south. Arid regions such as the Sahara or Arabian Peninsula

are clearly visible. Europe also exhibits quite low TCWV values. Synoptic features such as bands of elevated moisture are also visible. Despite the wide range of viewing zenith and solar zenith angles and their implications for the line of sight, we do not see any influence on the TCWV product. Over Central Africa we can see that some clouds, which are visible in the RGB, have not been detected by the cloud-mask. Such areas are also distinguishable by their decreased TCWV values compared to the surrounding areas. This underestimation due to clouds as well as the more fine details can be seen in a close-up of the scene in

Fig. 9b. Because of the 1 km resolution of FCI's NIR channels we can also detect meso- to mini-scale features such as smaller pockets of high moisture over the ITCZ or the mixing between dry and moist air masses. Under certain circumstances closer to the shore, the TCWV field shows slight discontinuities between the water and land-surface. The water-pixels very close to the shoreline often show values which deviate a few percent to the adjacent land-pixels, in most cases there is an over-estimation. This discontinuity is also visible in the TCWV field processed from OLCI and other NIR-TCWV retrievals.

At this stage, a rigorous quantitative validation of the TCWV product is not feasible and our comparison against TCWV from the ERA5 Reanalysis is not meant as such. Rather, as a preliminary way to check the TCWV field for consistency, we also plotted the relative difference between the FCI TCWV and a collocated ECMWF ERA5 Reanalysis TCWV, shown in Fig. 8c. This gives us a first impression if the algorithm works as intended and if there might appear some artifacts due to viewing- and or sun-geometry. The image in Fig. 8c is dominated by negative differences, which translates to a dry bias against the

Reanalysis TCWV. On average, FCI TCWV is approximately 10% drier than the TCWV of the Reanalysis over land surfaces and 5% drier over water surface. Furthermore there are areas with positive and negative differences close to one another, often resembling a line, i.e., over Northern Africa or over the South Atlantic. Fig. 8d depicts the averaging kernel at each pixel. Over land the value is close to 1 for most pixels since the forward model is very sensitive towards changes in the measurements. Over water this value lies between 0 and 0.7. In areas of light and or medium sun-glint, the averaging kernel ranges between

0.4 to 0.7 (very central glint area). In areas with low water-surface reflectance the averaging kernel approaches 0. Where there is a higher AMF or higher values of TCWV, the averaging kernel is slightly higher between 0.1 to 0.3.





MTG-FCI

MTG-FCI

(a) FCI red-green-blue composite.

(b) Retrieved FCI TCWV field.

S3A-OLCI

S3A-OLCI/SLSTR "FCI-alike"

(c) OLCI red-green-blue composite.

(d) Retrieved OLCI TCWV field.

**Figure 9.** Comparison of FCI TCWV and OLCI/SLSTR "FCI-alike" TCWV for a close-up on the 27 June 2023 over Northern Mali.

To showcase FCI's spatial resolution, we visually compare a TCWV field from Sentinel3-A OLCI/SLSTR with a zoomed-in
look of a TCWV field from FCI's test data in Fig. 9. Both are processed with the algorithm described above. The temporal
difference between the two fields is approximately 5 minutes. The scene is situated over northern Mali in West Africa. We can





see the differences in viewing geometry between FCI and OLCI. In the natural colour RGB of FCI, longer cloud shadows are visible which are much smaller in the S3A-OLCI image or their positions are shifted. The TCWV fields reveal moist air mass in the South-East, while a drier air mass is positioned in the North-West. Consistent with the comparison against the ERA5 analysis, FCI TCWV is about 10% lower than OLCI TCWV. Hence, another colourmap-range is used in the FCI TCWV

image(9b). FCI is capable to reproduce the amount of detail found in the OLCI TCWV field: e.g., a dry line in the western half of the image (i.e., strong gradients in moisture between the air masses) or gravity waves. The positioning of features appears to be coherent between the two sensors. Furthermore, we can see slight indications of the scan-lines in Fig. 9b. These are noisy pixels that follow lines which run from East to West. The effect is more pronounced over water surfaces.

Overall the level of detail both products exhibit is high and gravity waves in the lower atmosphere are captured (local peaks

and torughts in TCWV). In both figures, the effect of unidentified cloud pixels on the TCWV is visible: there, TCWV is underestimated and considerably drops compared to the surrounding clear-sky pixels. In contrast, there are some thin dust layers visible in the North-Western and Central-Eastern part of the RGBs which do not show through in either of the TCWV prodcuts.

## 5   Discussion and Outlook

In the validation against the reference ARM SGP TCWV dataset, the "FCI-like" OLCI 2-band TCWV shows a good performance with a bias of of 1.848 kg/m$^2$, RMSD of 2.235 kg/m$^2$, cRMSD of 1.256 kg/m$^2$and high $r^2$ of 0.99. The wet bias can be corrected following the procedure described in Preusker et al. (2021). In a comparison against their COWa algorithm applied to the same matchup dataset they have a similar R$^2$ of 0.99 but lower RMSD of 1.3 kg/m$^2$, which may well be attributed to both the use of an additional absorption band at 940 nm and initial $\tau_{pTOA}$-correction. The key performance indicators of the

two-band approach still show robust retrieval results.

Upon visual inspection of the daily, global composites of TCWV, it can be seen that our algorithm reproduces both large- and small scale patterns down to a resolution of 1.2 km very well. The global comparison against the reference networks returned slightly lower performance indicators with $r^2$ between 0.95 to 0.97, bias between 1.23 to 3.12 kg/m$^2$, MAPD between 6.98 and 41.243 %, RMSD between 2.55 to 3.91 kg/m$^2$and cRMSD between 1.88 to 2.35 kg/m$^2$. The highest RMSD and bias is

found in the comparison against AERONET, which is most likely due to AERONET's dry bias (Pérez-Ramírez et al., 2014). Comparing the OLCI/SLSTR matchups against the multi-year matchup of only OLCI in, We observe a lower performance. This is due to much fewer matchups during a shorter time span and high geographic spread.

For a rigorous validation, the assessment of a much longer time period will be essential. However, we can show that the PCR does not drastically reduce the algorithm's performance.

A crucial aspect that needs to be taken into account in the retrieval framework is the accurate estimation of surface reflectance in the absorption band. This is specific to the FCI instrument due to the absence of a second near-by window band (in contrast to OLCI). We implemented an approximation of the normalised radiance at the surface ($nL_{TOA}{}^*$) in the absorption band using a PCA regression. As shown in Figs. 3a and 3b The approximation shows a good performance against the next-best estimate,





i.e., extrapolation from two adjacent window bands using OLCI measurements and exceeds the performance of just using the

window band at 0.865 $\mu$m. Comparing $nL_{TOA}{}^{*PCR}$ against $nL_{TOA}{}^{*ext}$ reveals a bias of 0.001 $sr^{-1}$ and RMSD of 0.002 $sr^{-1}$. The approximated $nL_{TOA}{}^{*}$ may deviate from the "real" $nL_{TOA}{}^{*}$ on average by 1.5 % over land and 3.5 % over water. In rare cases the PCA regression failed.

The cases in which there is a very high deviation other than clouds mostly occur along rivers, coasts, in high elevations or at the poles. We assume that there may be several processes at play which require deeper investigations:

1.  water pixels with a complex composition (e.g. high loading of sediment or phytoplankton)

2.  sub-surface sea-floor reflections

3.  water pixels which actually contain sea-ice

4.  adjacency effects between surface interfaces (e.g. land-water, sea ice-water)

5.  snowy pixels which may be cloud-contaminated

6.  mis-alignment between OLCI/SLSTR, exacerbating all of the above

These effects may introduce over or underestimations of TCWV and are partially responsible for the discontinuities observable over lakes, rivers, shorelines. Discontinuities and overestimations of ocean pixels close to the shoreline or inland waters can also be observed in other NIR-TCWV products.

Such deviations introduce additional uncertainty in the initial estimate of $\tau_{pTOA}$ and may explain some of the increased

RMSD when compared to ARM using the extrapolated $nL_{TOA}{}^{*}$ in Fig. 5 and $nL_{TOA}{}^{*PCR}$ estimated from PCR in Fig. 7c. These types of mis-characterisations of the surface reflectance and the uncertainty of $\tau_{pTOA}$ translate to a further uncertainty of about 1 to 2 kg/m$^2$. Nevertheless, these initial results demonstrate that our approach is effective and advancing well towards an operational TCWV retrieval framework for FCI.

The actual performance of FCI TCWV may deviate from these verification results using reference TCWV datasets quite

substantially, since the spectral characteristics and calibration are different from OLCI. E.g., FCI's window band at 0.865 $\mu$m is much wider and contains some residual WV absorption. Future validation studies have to be conducted for further characterization, which may also lead to a more elaborate correction for initial $\tau_{pTOA}$ estimation. To assess the functionality of the current algorithm prototype, we applied it to the mtgt505 FCI L1c test dataset provided by EUMETSAT. These are real FCI data captured on the 13th of January 2024. Conceptually, everything is in working order. The running times are close

to or below the 5 minute mark (FCI's nominal temporal resolution on a 2024 computer) and allow for a near-real-time and operational application of our TCWV product. For a fair comparison and accurate interpretation of TCWV field results, a good FCI cloud mask is essential.

Full-disk comparisons show that the algorithm produces a sensible TCWV field. The relative difference between collocated ECMWF ERA5 Reanalysis TCWV at 12 UTC and our TCWV product reveal that the majority of the field shows negative

deviations, indicating a predominant dry bias. Furthermore, there are large-scale patterns of positive and negative deviations close to one another. Such patterns are to be expected in a comparison against model data and indicate that the model struggles



with accurately capturing the advection of air masses in both space and time, while the observed TCWV fields might be closer to the actual state.

Using this initial comparison against ERA5 analysis we can deduce a systematic dry bias of approximately 8%. We see three probable reasons for this systematic dry bias: 1) the bias might be related to the preliminary calibration of the FCI data, 2) the PCR systematically over-estimates the surface reflectance at 0.914 $\mu$m and thus $\tau$ is too low and 3) undetected deficits in our LUTs. If this systematic bias persists in a future validation study we may mitigate it using Preusker et al. (2021).

The analysis of the averaging kernel, a related OE output parameter, reveals by how far the retrieved state is influenced by the measurement. For FCI TCWV, over land this value is close to 1, indicating a high sensitivity to the measurement and only a very small contribution of the prior knowledge. This is also referred to as the algorithm being independent from the NWP input. This property is a key advantage of the TCWV retrieved from the NIR in contrast to other satellite-retrieved sources of TCWV, which may either heavily depend on NWP inputs or provide a much coarser resolution. Over the majority of water surfaces, the averaging kernel is lower. This is mostly due to the much lower surface reflectance of the water surface resulting in a lower signal at the sensor. In some cases the surface reflectance is close to 0, making a retrieval of TCWV by differential absorption very challenging. However, the OE still provides an update of the prior TCWV field. In such areas, the averaging kernel is close to 0. Over sun-glint dominated areas with higher surface reflectances, the averaging kernel lies between 0.4 and 0.7, allowing for more accurate and independent TCWV retrievals in these regions.

Comparing OLCI and FCI TCWV up close we can easily see that FCI TCWV matches the level of detail found in the OLCI TCWV product. Patterns are well reproduced and areas affected by north-eastern winds are actually shifted towards the west, while patterns in the southern part of the image, where no strong winds can be found, remain stationary. For scenes over Europe, the resolution compared to OLCI will be slightly decreased due to the curvature of the Earth.

The considerable under-estimation of TCWV in the presence of clouds is due to not-accounted for shortening of the photon paths and 3D effects. As a first measure, a better cloud mask is needed to filter out such pixels. At a later stage, such retrieved pixels may be used for an "above cloud" water vapour product. Such a product may then later be used for the detection of WV entrainment into the stratosphere, e.g. in the presence of overshooting tops (Setvák et al., 2008; Dauhut et al., 2018; Khordakova et al., 2022). The stripes of enhanced noise which run across the FCI TCWV image are caused by scan-lines of FCI. Similar scan-line artifacts are found in whisk-broom sensors, too, such as MODIS or VIIRS. Over land this effect is not that pronounced, however, over dark water pixels it is more noticeable. This may change in future Level 1c processing versions.

The assessment exercises discussed above helped us identify several limitations and challenges regarding TCWV retrievals from FCI measurements. One challenging factor is the spectral extrapolation using a reference calculated from the PCR. This challenge may be addressed by 1) extending the training reflectance spectra dataset the PCA is built from with exemplary spectra from different water bodies (i.e. fresh-water, chlorophyll dominated, sediment dominated, etc.) or 2) a dedicated water PCA or 3) a dedicated inland-water LUT, containing all the aforementioned parameters. Another fundamental issue is that over water surfaces the inversion framework is under-determined: a measurement vector with only two elements $(nL_{TOA}(0.865), \tau_{pTOA}(0.914\mu m))$ is opposed by a state vector with at least three elements (TCWV, AOT, WSP). Outside of sun-glint areas the influence of the wind speed is marginal and AOT mainly increases the TOA signal (and thus the forward





model is not sensitive to changes of the windspeed) and inside sun-glint the surface signal is much stronger and the influence of a thin layer of aerosol reduced (similar to land surfaces). Because of that, the information-content is relatively balanced and the impact slightly reduced. However, adding an additional third channel to the water-part of the algorithm (e.g. 0.550 $\mu$m or
1.61 $\mu$m ) may improve performance.

Our framework may be adapted to provide accurate TCWV retrievals for other sensors featuring at least two channels in and around the $\rho\sigma\tau$ band. The National Oceanic and Atmospheric Administration (NOAA) is commissioning GeoXO Imager (GXI), the successor to the Advanced Baseline Imager (ABI) on the instruments on the Geostationary Operational Environmental Satellite - 3rd generation (GOES) which include a WV absorption band in the NIR (Lindsey et al., 2024).
Another future instrument soon to be launched into a polar orbit is METImage, flying onboard EUMETSAT's Meteorological Operational satellite second generation A (METOP-SG-A) (Phillips et al., 2016). METImage will enable NIR TCWV with a spatial resolution of 500 m and global coverage every day. METImage will also provide O2A band measurements which can be used to reduce ambiguity due to shielding of cirrus or elevated aerosol layers. A NIR TCWV product from METImage may then be used in advanced synergies with sounders such as Infrared Atmospheric Sounder Interferometer - New Generation
(IASI-NG) which will also be flying on METOP-SG-A. IASI-NG is the successor of IASI which provides all-sky temperature and humidity profiles with a slightly lower accuracy in the presence of clouds (Müller, 2017).

Furthermore, the Infrared Sounder (IRS) will be operating on MTG-S1, MTG-I1's sister satellite, which covers the same field of view as MTG-FCI. This will enable a synergy between TCWV from FCI and the IRS humidity profile product. NIR TCWV could very well complement profile soundings for both IASI-NG and IRS: one short-coming of these retrievals is their
low or missing sensitivity to the lowest layers of the troposphere (below 1-2 km). Furthermore, their spatial resolution is in the order of tens of km, often insufficient in assessing small-scale weather patterns. A high-spatial resolution NIR TCWV product, sensitive to the whole column of air could complement such sounding products perfectly, albeit in the absence of clouds. A synergy could consist of an updated layer-product or a product which provides the moisture content of the lowest levels of the troposphere. Such synergy products could provide crucial insights of the meteorological conditions such as the atmospheric
instability and improve the potential for the prediction of severe weather.

Yet, a more zoomed-in look into a FCI TCWV field obtained from real data reveals that FCI TCWV alone is capable of observing small-scale patterns also visible in an OLCI/SLSTR TCWV field. The capability of seeing such small-scale WV features (e.g. convergence zones, convective rolls, deepening boundary layers) emphasizes the potential of an operational FCI NIR TCWV product for nowcasting purposes. Last but not least, and to further highlight the potential of MTG-FCI based WV
observations for convective nowcasting purposes, we showcase the TCWV field from Fig. 9b again together with the TCWV from two time steps later in Fig. 10a to 10c. The sequence demonstrates, how one can track the propagation of the gravity waves and the north-western movement of the moist airmass along the moisture-front. What is more compelling, is, that we can detect the formation of small updrafts or thermals indicated by the stark increases in TCWV from 10b to 10c. This results in a pattern similar to convective rolls shown in Carbajal Henken et al. (2015). In the lower center, at around 11:40 UTC first
clouds are forming.




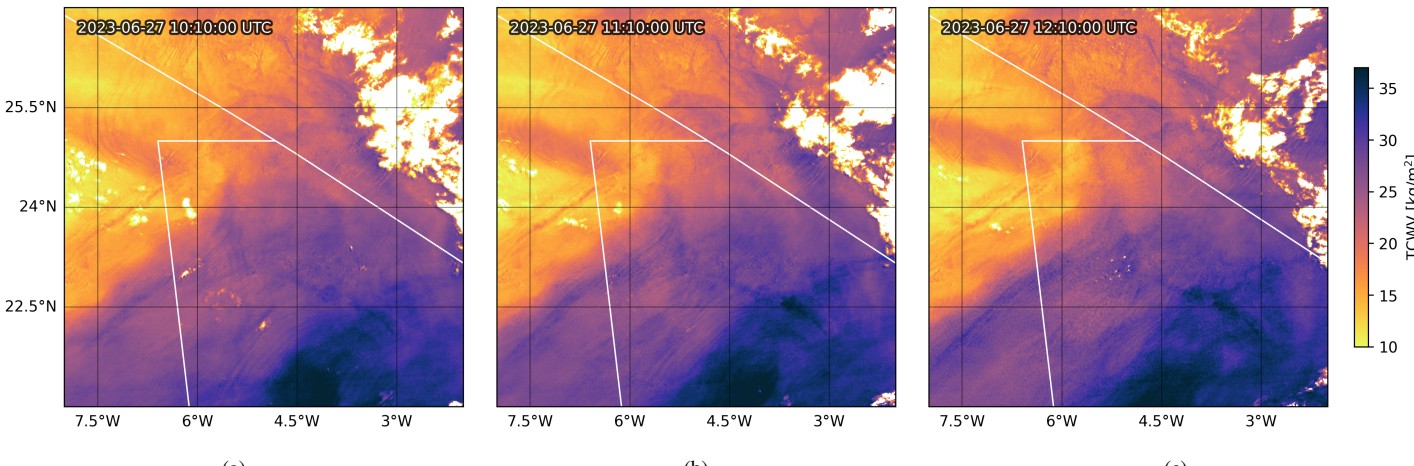

**Figure 10.** Time sequence of FCI TCWV shown in Fig. 9b with 1 hour between each frame.

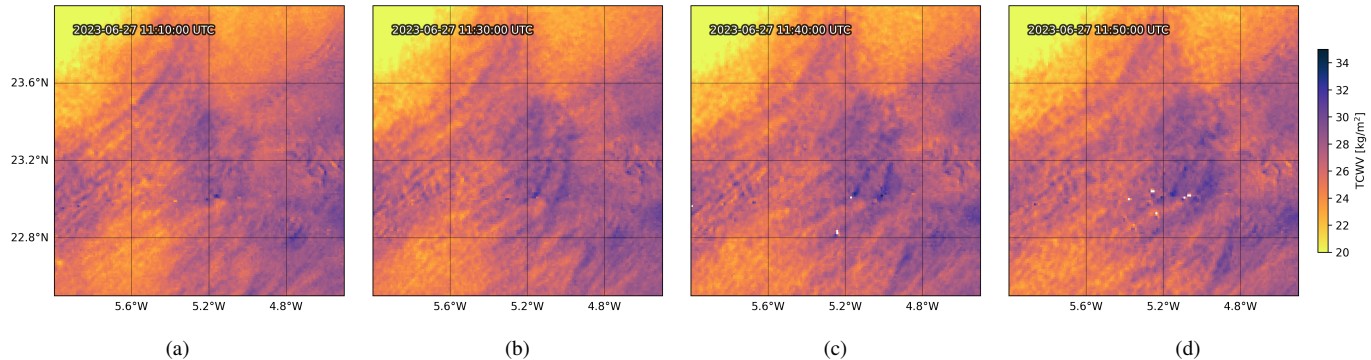

**Figure 11.** Time sequence of FCI TCWV shown in Fig. 11b to 11c with 10 minutes between each frame, leading up to cloud formation at 11:40 UTC. The range of the colormap has been further adjusted to show more detail. The cloud formation is visible in RGBs (not shown) and be distinguished by the sudden appearance of clouds (i.e., white spots) in 11c to 11d.

An even closer look at the time shortly before these clouds forming, FCI NIR TCWV reveals even more intricate details in Fig. 11. The pattern structure shows what are assumed to be individual convective cells growing in depth, shortly before cloud formation.

This case highlights both the remarkable large-scale and small-scale WV features that will be observable with FCI. FCI 620 will not only provide a snapshot of convective cloud formation but a steady stream of data. These show how the TCWV field changes in clear-sky regions around clouds or before the onset of cloud formation, given the presence of sufficient daylight. With real FCI data we can monitor the temporal evolution of TCWV fields, allowing us to track and comprehend both large-scale and small-scale dynamics before, during and after convective development. Images such as Fig. 11 could be provided by MTG-FCI way in advance of the formation of the first clouds and could help to pin-point areas of local convergence and





increased low level moisture for nowcasters. Above that, the spatio-temporal resolution of MTG-FCI may even provide deeper insight into the conditions of severe storm formation.

## 6 Conclusions

Leveraging our expertise in Total Column Water Vapour (TCWV) retrievals from Near-Infrared (NIR) measurements for various satellite-based passive imagers, we developed a new retrieval framework for the new Meteosat Third Generation Flexible
Combined Imager (MTG-FCI) measurements. Although the latest calibrated Level 1 FCI data were not available by the time this work was completed, we utilized both OLCI data and an initial version of L1 MTG-FCI data, alongside reference TCWV data from other sources, to assess the performance of this two-band NIR-TCWV retrieval framework. The evaluation exercises highlight the robustness of the retrieval framework and helped identify specific challenges and limitations related to the MTG-FCI instrument, which can be further addressed with fully calibrated FCI data in the near future.

As the successor to MSG-SEVIRI, MTG-FCI boasts extended observational and spectral capabilities that promise significant advancements in weather and climate research and applications, particularly in the monitoring and study of atmospheric TCWV amounts and dynamics. Notably, FCI is the first geostationary satellite instrument with measurements in the NIR $\rho\sigma\tau$ WV absorption band. While SEVIRI TIR measurements allowed to derive information on WV amounts mainly in higher parts of the troposphere, the FCI NIR WV absorption measurements exhibit the greatest sensitivity to WV amounts near the surface. This
enables accurate and high temporal resolution observations of changes in moisture content in the lower troposphere. Consequently, these novel and comprehensive TCWV observations will enhance the (real-time) monitoring of atmospheric moisture distributions in the boundary layer, their evolution and associated meteorological phenomena across regional to continental scales, with the potential to significantly advance nowcasting techniques.

*Code availability.* After further refinement the code will be made available via the NWCSAF GEO software package via: https://www.nwcs
af.org/nwc/geo-geostationary-near-real-time-v2021

*Data availability.* OLCI/SLSTR data obtained from here: EUMETSAT. ERA5 data obtained from here: Service and Store (2023). Satellite AOT obtained from Copernicus Climate Change Service and Climate Data Store (2019)

*Author contributions.* Conceptualization, J.E.K., C.C.H., R.P.; methodology, J.E.K., C.C.H., R.P.; software, R.P., J.E.K.; validation, J.E.K.; formal analysis, J.E.K., C.C.H., R.P.; investigation, J.E.K., C.C.H., R.P.; resources, J.E.K., C.C.H., R.P., X.C., J.F.; data curation, J.E.K.;
writing—original draft preparation, J.E.K.; writing—review and editing, J.E.K., C.C.H., R.P., X.C., P.R.; visualization, J.E.K.; supervision, R.P.; project administration, X.C., P.R.; funding acquisition, J.E.K., C.C.H., R.P., P.R., J.F..



*Competing interests.* The authors declare no conflict of interest.

*Acknowledgements.* This research has been partially funded by the Satellite Application Facility in Support to Nowcasting and Very Short Range Forecasting (NWCSAF) of the European Organisation for the Exploitation of Meteorological Satellites (EUMETSAT) through the
Associated Visiting Activity NWC_AVS23_01, by EUMETSAT COWa Contract EUM/CO/18/4600002115/EJK, by EUMETSAT FRAME Contract EUM/CO/24/4600002869/JoSt and by the Deutsche Forschungsgemeinschaft (DFG, German Research Foundation) - Project number: 320397309 (TP2 QPN) within FOR 2589 "Near-Realtime Quantitative Precipitation Estimation and Prediction" (RealPEP). We acknowledge the legacy and software of COWa that went into this algorithm supported by the Remote Sensing Products (RSP) division of EUMETSAT. We thank NWCSAF and EUMETSAT for the provision of preliminary MTG-FCI data. We thank all researchers and staff for
establishing and maintaining the AERONET Sites, SUOMINET Sites and ARM Sites used in this investigation.



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
