# Peer review of "Optimal Estimation Retrieval Framework for Daytime Clear-Sky Total Column Water Vapour from MTG-FCI Near-Infrared Measurements"

_EGUsphere, 2024_

## Author Response (AR1)

**Response to Review 1**

**In this paper, the authors develop a relatively simple yet powerful total column water vapor retrieval targeting the 0.91 micrometer solar water vapor absorption band that exists on several satellite sensors, most notably the MTG FCI. Its 1 km spatial resolution and 10 minute refresh will allow an unprecedented ability to track low-level water vapor. The scientific quality of the paper is excellent and I recommend it be accepted following only minor revisions, mainly related to presentation refinements.**

*Thank you for your very positive evaluation of our manuscript and your valuable comments. They will certainly improve the quality of our study.*

*Please note that a second reviewer requested a major revision, specifically recommending a stronger focus on MTG-FCI data and improvements in structure and language. Therefore, we conducted a comprehensive revision of the manuscript. We removed redundancies, shortened overly long sections, and streamlined the text, particularly in the Introduction, Physical Background, Inversion Method, Result and Discussion sections.*

*Additionally, we introduced a new, concise section titled "Finalized Retrieval Framework", which provides an overview of the retrieval processing chain. We also clarified and emphasized the reason for using OLCI/SLSTR synergy data to set up and evaluate the retrieval framework, with its specific challenges like estimating surface reflectance in the absorption channel, for the new MTG FCI data.*

*Furthermore the matchup database against reference TCWV observations was extended and covers more dates of 2021 and more sites.*

**Minor comments:**

**1)  Line 38 and throughout the paper: the authors refer to "ρστ water vapor (WV) absorption regions" - I'm personally not familiar with the use of "ρστ" in this context, so it's worth introducing the meaning or definition here.**

*Yes, indeed. The origin of this "ρστ" lies in Samual P. Langley's observations of dips in radiation emitted by the sun which he assigned latin/greek letters. For some applications these names are still in use today. I will clarify this in the text, as I had to look for this background information myself.*

*Changes to the manuscript, lines 48-540:*

*The use of the so-called ρστ WV absorption region in the NIR (0.9 to 1.0 μm) is not new. This designation stems from observations of atmospheric absorption of solar radiation in the 19th century (Langley, 1902). There, light is much more likely to be absorbed by WV molecules compared to spectral regions outside these absorption features (window regions)*

**2) Lines 45-47: I suggest mentioning that IR techniques including the split-window difference depend on the atmospheric temperature profile (in addition to the water vapor profile), making it more complicated. In other words, using only the 0.86 and 0.91 channels [largely] avoids this temperature-related complication.**

Good suggestion. We will add this detail in the text.

Changes to the manuscript, lines 45-52:

*On the one hand, a split-window technique using weakly absorbing WV measurements can be employed to retrieve TCWV or boundary layer WV with relatively high uncertainties (e.g., Kleespies and McMillin, 1990; Casadio et al., 2016; Hu et al., 2019; Dostalek et al., 2021; El Kassar et al., 2021). Lindsey et al. (2014) and Lindsey et al. (2018) showed that the split-window difference by itself may already provide valuable insight on the WV content in the boundary layer or lowest layers of the troposphere. On the other hand, measurements from strongly absorbing WV bands serve to retrieve WV amounts limited to upper tropospheric levels and/or layered WV products (e.g., Koenig and De Coning, 2009; Martinez et al., 2022). However, due to the absorption and re-emission of radiation by WV in the IR, such approaches rely on knowledge of the atmospheric temperature profile in addition to the atmospheric WV profile. Using observations in the VIS/NIR largely avoids these temperature-related complications.*

*I also added a further reference to another TCWV retrieval exploiting the split-window (Advanced Infra-Red WAter Vapour Estimator (AIRWAVE) algorithm).*

**3) There are a number of minor English grammatical errors throughout; one example is in line 56, "has been initiated" should be "has initiated." Instead of pointing out all of them, I suggest doing a thorough read-through before resubmitting.**

*Thank you, we will re-read the manuscript thoroughly.*

**4) Line 65: use of the word "disturbing" here is odd. I know what you're trying to say, but perhaps rephrase.**

*We changed the wording as follows, lines 38 – 40:*

*Apart from that, WV is considered an inconvenient atmospheric component for several remote sensing applications, e.g., surface parameter retrievals, for which precise information on WV amounts in the atmosphere is needed for atmospheric correction methods (e.g. Gao et al., 2009; Wiegner and Gasteiger, 2015; Valdés et al., 2021).*

**5) Line 121: "was" instead of "has been".**

*Changed.*

**6) Line 134: Don't reference an equation (13) here that appears later in the paper. Either introduce the equation here or save its mention for later.**

Removed that reference to the equation since it was not strictly necessary.

**7) Line 167 and throughout: I believe it's more standard to reference UTC time using the 24 hour clock instead of AM and PM. So 6 am UTC should be 0600 UTC and 6 PM UTC should be 1800 UTC.**

Good point, we changed it accordingly.

**8) Line 180: An interesting experiment (perhaps mentioned at the end as future work) would be to assess the sensitivity of the results to changes in AOT.**

Yes, to some extent this has already been studied (e.g., Diedrich2015 future instruments). As mentioned in the paper, over bright surfaces, the effect is negligible. However, both the composition, amount and layer height are important to consider. We added the citation accordingly in line 217 in the restructured "Physical Background" section.

**9) Figure 1 caption: isn't that line red and not green in the bottom panel?**

*Correct, in an earlier version of the figure there was a green and red line. We corrected the caption.*

**10) Line 239: Related to my comment above about temperature sensitivity, here you mention that pressure-dependent line broadening may play a role. How large of a role? Non-negligible? As a reader I'm curious.**

*Very good point. The influence of pressure broadening on average is larger than the effect of temperature broadening. It is difficult to quantify one without the other. In many cases the effect is negligible, however, in specific cases (e.g., very dry, very cold or very low pressure) the effect is non-negligible.*

*Lindstrot et al. (2012) tested for the error introduced by simplified (wrong) temperature and pressure assumptions. They found that having a single average temperature profile RMSD 1.3 kg/m², bias of 0.6 kg/m² and a mixture of 2 different temperature profiles removed the bias and reduced the RMSD to 0.3 kg/m² when compared to the TCWV calculated using the more accurate temperature profile retrieved from NWP.*

*In another test, they compared the effect of using a constant surface pressure (1013 hPa) against using the surface pressure provided by NWP and such a simplification results in an RMSD of 0.9 kg/m² and bias of 0.5 kg/m² with the largest deviations found in mountainous regions.*

*We changed the mentioned passage in the text accordingly (lines 226-230):*

*Over both land and water surfaces the atmospheric temperature profile and surface pressure play a lesser role due to temperature- and pressure-dependent line broadening (Rothman et al., 1998). In contrast to TCWV retrievals in the IR, the impact of the temperature profile is substantially lower but not negligible. The uncertainties due to a mis-characterised temperature profile are approximately 0.6 kg/m² and surface pressure at about 0.9 kg/m² (Lindstrot et al., 2012).*

**11) Line 258: following equation 3, it would be easier to follow if you define each of its terms in the same paragraph directly below the equation, instead of extending those definitions into the subsequent paragraph.**

*Good point. The structure was indeed a bit confusing. We cleaned up the text and structured the equations and terms a bit more straightforward. The changes in line 255-272 are:*

*Equation 3 [...]*

*where AMF is the air mass factor, nLTOA\* is the normalised radiance corrected for the influence of WV absorption, a and b are the so-called correction coefficients which may correct for a systematic bias discovered in a validation against reference TCWV observations.*

*The AMF is calculated as follows:*

*Equation 4 AMF= [...]*

*Dividing through √AMF, the relationship between TCWV and τpTOA becomes more linear, reducing the number of necessary iterations in the inversion later on. nLTOA\* needs to be approximated using other available information (e.g., a climatology atlas, neighbouring window channels). Here, we use a more elaborate technique, described in Subsection 3.4. Preusker et al. (2021) have obtained the correction coefficients a and b by minimizing the differences between simulated and measured OLCI observations using ARM-SGP.C1-MWR TCWV as an input (see Preusker et al. (2021) for details). For OLCI's version of this algorithm, a and b for band 19 (at 0.9 μm) were estimated to be -0.008 and 0.984, respectively, from the results shown in Sec. 4.1. For FCI, other MWR TCWV references will be necessary. We intend to use reference sites such as Meteorological Observatory Lindenberg – Richard Assmann Observatory (MOL–RAO) (Knist et al., 2022), the Cabauw Experimental Site for Atmospheric Research (CESAR) (Van Ulden and Wieringa, 1996) or ARM — Eastern North Atlantic (ENA) (Mather and Voyles, 2013).*

**12) Equation 4: Make AMF either caps or not caps (amf) consistently.**

*Changed.*

**13) Lines 285-302: Has this same derivation been presented in one of the referenced papers? If yes, there's no reason to recreate it here. This is a good opportunity to shorten a relatively long paper.**

*Good point, in fact is has been presented in two of the referenced paper and we will shorten it!*

**14) Line 351: I don't know the word eigenwert? Is that in German?**

*Yes, the word is German and was coined by David Hilbert (1904). There is another term called Eigenvalue or "characteristic value". We changed Eigenwert to Eigenvalue.*

**15) Figure 3 captions (and later figure captions): maybe this is a requirement for this journal (?), but it's unusual to define the subparts of the figures separately from the figure caption itself. Usually the subparts are defined within the single figure caption.**

*To be honest, I am not really sure. I thought this would be a logical way to separate the distinct parts of the subfigures and only mention the common parts. We will let the editors decide on how to proceed.*

**16) Figure 7c) caption: I don't think these dots are coloured with relative frequency of occurrence.**

*Correct, we corrected the caption.*

**17) Line 446 and afterward: I suggest using the term "true color" instead of "natural color" to describe these red/green/blue images. Natural color has a different meaning, often.**

*Good point, we changed the text accordingly.*

**18) Regarding Fig. 9: this falls into the future work category, but it would be really interesting to compare Fig 9b to a version that uses only IR channels, both to assess the improved spatial resolution and to see whether there are low-level WV features that the IR retrieval misses.**

*This is a really good point. We consider to include this in the follow-up paper.*

**19) The Discussion and Outlook section is very good, but perhaps a little too long. This is really up to the journal editor.**

*Our aim was to underpin the value and possibilities of this retrieval without holding back the challenges and limitations. In response to the other reviewer's and your comment we shortened and restructured the discussion to be more concise.*

*In summary, this paper is a very important contribution to the literature. I'm very excited about the prospect of the retrieval being optimized using more real FCI data, then being incorporated into the NWCSAF GEO software for real-time, operational use. It has the potential to be a game-changing tool for use by forecasters in detecting and tracking low-level water vapor between clouds.*

**Response to Review 2**

**Summary**

The manuscript by El Kassar et al. presents a retrieval framework for total column water vapour (TCWV) in the near-infrared (NIR) spectral range, designed for future observations from the MTG FCI instrument. Due to limited availability of real FCI data, the authors develop a forward model to emulate FCI-like radiances using OLCI/SLSTR observations. This synthetic dataset is used to apply and validate the retrieval framework. The results show good agreement between retrieved TCWV and reference datasets. In addition, the authors apply the method to a first set of FCI test data, demonstrating initial promising results.

**General Assessment**

While the topic is relevant and the core idea has potential, the manuscript in its current form is far from ready for publication. It suffers from significant weaknesses in structure, clarity, language, and scientific rigour. In particular:

- The writing is verbose and difficult to follow, lacking a clear narrative structure.
- The English is poor, with frequent language issues and awkward phrasing.
- Some figures are incorrect or misleading (e.g. missing or mislabelled elements).
- The presentation gives the impression of a hastily compiled manuscript, which undermines the quality of the work.

As a result, I would place this submission on the borderline between major revision and rejection. Nevertheless, I believe the methodological idea is interesting and worth pursuing. I therefore recommend a **comprehensive major revision**, with the expectation of a thoroughly rewritten manuscript.

We thank the reviewer for their thorough evaluation of our manuscript. We appreciate the recognition of the methodological potential and relevance of our work, and we take the concerns seriously. Below, we address the key points raised.

- We acknowledge that the manuscript in its current form may lack clarity in language and structure for some parts. A full revision of the manuscript has been performed, refining text and presentation and reducing redundancies.  More specifically, most significant updates regarding this issue were done as follows:

    - Introduction Section: Restructured to emphasize water vapor importance, review current methods, and introduce NIR approaches leading to the MTG FCI focus

- Data Section: Streamlined by shortening lengthy parts and removing redundancies.

- Physical Background Section: Made the section more concise with a bit of restructuring to enhance flow.

- Inversion Section: Reduced length by removing detailed standard Optimal Estimation (OE) equations, instead referring to key references for the methodology. There was a mix-up between number of maximum iterations, it now correctly states that the max. number of iterations is 6 over land and 8 over water surfaces.

- New Section "Finalised Retrieval Framework": Added a reorganized section providing a clear overview of the entire processing chain without introducing new text.

- Results Section: We chose to exclude the global plot of OLCI-COWA and OLCI/SLSTR TCWV from the results. We extended the OLCI/SLSTR "FCI-like" TCWV matchup dataset to cover all months of 2021 and also included the CONUS Suominet GNNS sites. This drastically increased the number of data points and statistical robustness. We also filtered out elevated stations with elevations over 3 km as some of these showed systematic offsets. These datapoints did not significantly impact the metrics (RMSD, bias , etc.).

- All figures/tables are carefully reviewed and corrected where needed.

Moreover, as actual FCI L1c data are now available, I strongly encourage the authors to include retrieval results based on real observations, rather than relying exclusively on emulated or synthetic data. The concept of generating synthetic FCI-like radiances is still valuable, and the corresponding methodology and validation results could be retained as supporting material — for example, in an appendix or the supplementary information. This would allow the manuscript to focus more clearly on the application and evaluation of the retrieval framework under realistic conditions.

The NIR-based TCWV retrieval framework can build on work done for NIR WV absorption channels and nearby window channels for OLCI and MODIS (Cowa retrieval). This retrieval framework can be applied to new and future satellite missions that share similar band settings, like FCI, MetImage, GEOX, LSTM (for which FCI is the first in line and the current focus of our work). What FCI and these future sensors have in common is that they have more limited spectral sampling in this spectral region compared to OLCI/MODIS, which introduces specific challenges, particularly for modeling surface reflectance in the absorption channels.  At the time of the core part of this work, fully calibrated FCI Level 1 data were not yet available and only became accessible end of 2024 (note that FCI L1 data are being refined via external calibration methods, the EUMETSATs MICMICS system, after

an on-board calibration mechanism issue). However, to anticipate and address these challenges, we used the opportunity to use the OLCI/SLSTR synergy to establish an adapted retrieval framework accordingly. We call this "FCI-like" data, which are not synthetic data. Additionally, OLCI has well-known radiometric characterization, worldwide coverage and long-term record, and we consider this approach as a practical and robust basis to test performance and refine the retrieval framework under a wide range of realistic conditions.

In response to the reviewer's suggestion, we have now clarified this motivation more explicitly in the manuscript  (emphasized in abstract, introduction and conclusion).

Recently, we have started applying the retrieval framework to real FCI L1 data. These efforts are currently ongoing and include extensive validation. Given the dense reference TCWV observations in Europe, FCI data for summer months will be helpful to cover the very moist conditions as well. The results will be presented in a dedicated follow-up publication that focuses specifically on retrievals based on real FCI observations, covering extended time periods and geographical domains, as well as analyzing implications for applications like usage in nowcasting.

**Specific Comments**

Note: This review focuses on content and structure. Issues with language, spelling, and phrasing are not addressed in detail here, but are pervasive and require careful editing.

Introduction

- Needs reorganization: start by motivating the importance of water vapour retrievals, then introduce NIR capabilities, followed by the role of MTG/FCI (especially for nowcasting).
- Reduce length and focus on core arguments.
    - We reorganized and streamlined the Introduction as suggested

Data section (Section 2)

- ERA5 is not a forecast product – it is a reanalysis. Please correct this terminology.
    - We have clarified in the text. ERA5 renalysis provides also a forecast product (not an operational one). We refer to the ECMWF ERA5 documentation for more details: https://confluence.ecmwf.int/pages/viewpage.action?pageId=85402030
- Why is ERA5 used at 3-hour resolution? Why not use ECMWF IFS forecasts directly, especially since you plan to use them in the future?
    - In an operational context, we plan to use short-range ECMWF forecasts. However, for this study, the forecast fields from the ERA5 reanalysis provides a stable, consistent dataset to evaluate retrieval performance. Over land, where

averaging kernel values are generally very high, the prior has limited influence on the retrieval, making this ERA5 product a suitable stand-in.

- Why rely on a fixed aerosol climatology? What would happen during dust events (e.g. Saharan dust transport to Europe)? Consider using aerosol forecasts (e.g. from IFS) to improve consistency.
    - This is an important point. Currently, we use a fixed climatology, but acknowledge that dynamical aerosol inputs, especially during events like Saharan dust outbreaks, could improve retrieval performance under such aerosol conditions. We will better clarify the masking strategy and shortly discuss the potential benefits of including aerosol forecasts in future implementations.

- Radiosonde validation: Why are only ARM SGP data used? Why not include GRUAN, ARSA, or IGS GNSS data for more robust validation? Also: AERONET has known dry biases – why use it at all?
    - We use ARM SGP data due to the temporal coverage (matchup with OLCI) and known very good accuracy. The temporal matchup (<30 minutes) of GRUAN and ARSA with satellite pixels is limited to non-existent and spatial drift during radiosonde ascents introduces additional uncertainty.
    - AERONET data has the advantage of a very good geographical distribution and strict cloud filtering and continuous measurements. We are aware of its known dry biases and is mentioned in the text.
    - The inclusion of SUOMINET provides a broad global network of GNSS-based TCWV.
    - All three datasets showed consistent results, and we believe that additional datasets like IGS-GNSS would not change the evaluation and our conclusions.

- Clearly state error assumptions for the ODR method.
    - We did not include the uncertainties in the orthogonal distance regression computation, all data points have equal weights.
- Define "centered RMSD"; this metric may not be familiar to all readers.
    - We included a clearer definition for this metric.

Physical Background (Section 3.1):

- How is the water vapour continuum treated? It is not discussed but can be significant.
    - The simulations for this part (with CGASA and MT-CKD, reference to the MOMO RTD now included: https://www.sciencedirect.com/science/article/pii/S0022407314001447) used

in this study include the latest HITRAN database as well as treatments for self-broadening, far-wing continuum absorption, and line-mixing effects.

- Sentences are repeated verbatim – please check for redundant phrasing.
    - We carefully revised this section and removed any redundancies.

Forward Model (Section 3.2):

- Equation (3): Where does the sqrt(AMF) term come from? And what is the source of the input for nL_TOA* ?
    - This term is a practical solution to linearize the relationship and reduce the the number of iterations and the LUT sampling. This approach has proven effective in previous work (e.g., COWA). We would be happy to engage in a discussion if the reviewer is interested.

- Consider using a table of variable names for clarity.
    - *A good suggestion. We added a table listing all variables for clarity.*

- Tables 1 & 2: What are the parameter increments in the LUT? Regarding AOT: What about near-surface aerosol layers (e.g. at 0 m), or elevated layers (2–4 km)? What about different aerosol layer thicknesses?
    - We appreciate this good and interesting point. The current LUT design assumes fixed layer heights and thicknesses, but we agree that sensitivity to near-surface or elevated aerosol layers is an important topic and will include this in the outlook. It has been previously discussed in Diedrich et al. (2013), e.g., the challenges over dark surfaces.

- AMF depends on scattering and BRDF effects – how are these accounted for?
    - Our approach uses geometric AMF, which is not influenced by scattering or BRDF effects. The path length is computed in the RT simulations.

Inversion (Section 3.3):

- How are sun glint conditions over water handled?
    - Sun glint is included as a component of the forward model and part of the state vector. We acknowledge ambiguities between aerosol and glint effects, particularly in the region between high and low glint. We see this is reflected in reduced averaging kernel values in such conditions.

- Again, ERA5 is not a forecast. Please revise terminology.

OK

Section 3.4:

Claim that ρTOA=ρBOA needs justification. This neglects effects of broadband absorption from aerosols, the water vapour continuum, etc.

- The previous wording was inaccurate. We now state (TBD check exact wording):

"The underlying assumption is that between 0.865 and 0.914 µm, atmospheric scattering and attenuation other than WV are nearly identical. Thus, the ratio $\rho_{TOA}$(0.914 µm) / $\rho_{TOA}$(0.865µm) approximates the ratio ALB(0.914 µm) / ALB(0.865 µm) ."

Validation (Section 4.1)

- As above, the validation dataset is too limited. Why only a few AERONET and SuomiNet stations?
- Statistical robustness is lacking. Include additional sources like GRUAN, ARSA, IGS-GNSS, etc.
    - We have extended the time period and number of AERONET and SUOMINET stations included in the analysis to increase spatial and temporal coverage and statistical robustness. As mentioned above, the other datasets have limited temporal and spatial overlap.

Section 4.2:

- Define clearly how relative difference is calculated.
    - We now clearly defined this in the text.
- Consider including a brief explanation of averaging kernels, especially in the OE context.
    - We have shortened the OE section. We now define the scalar AVK used in this context, derived from $d\hat{x}/dx$. Other OE diagnostic parameters like cost and information content are also briefly explained.

Discussion (Section 5): the discussion is overly long and needs to be drastically shortened. Please condense.

We have reviewed this section and removed some redundancies and focus on the main interpretation points and implications for retrieval improvements and applications.

Figures

- Figure 1: The green line is missing.
- Figure 2: also wrong colors (METImage?) and wrong wavelength unit

We have corrected this.

References

Check reference formatting carefully – several entries are inconsistent or incorrect (e.g., El Kassar, EUMETSAT, Copernicus data, etc.).

We have checked this.

**Recommendation**

I recommend major revision, bordering on rejection. However, I encourage the authors to revise thoroughly and resubmit, with the following in mind:

- A full rewrite of the manuscript for clarity, structure, and language quality.
- A stronger focus on FCI as the target instrument.
- Inclusion of results using actual FCI observations, where available.
- Improved use of validation data and error characterization.

With serious revision, the study has the potential to make a valuable contribution.

**Citation**: https://doi.org/10.5194/egusphere-2024-3605-RC2